# UNDERSTANDING OF SERVER-ASSISTED FEDERATED LEARNING WITH INCOMPLETE CLIENT PARTICIPATION

## ABSTRACT

Existing works in federated learning (FL) often assumes an ideal system with either full client or uniformly distributed client participation. However, in practice, it has been observed that some clients may never participate in FL training (aka incomplete client participation) due to a myriad of system heterogeneity factors. To mitigate impacts of incomplete client participation, a popular approach is the server-assisted federated learning (SA-FL) framework, where the server is equipped with an auxiliary dataset. However, despite the fact that SA-FL has been empirically shown to be effective in addressing the incomplete client participation problem, there remains a lack of theoretical understanding for SA-FL. Meanwhile, the ramifications of incomplete client participation in conventional FL is also poorly understood. These theoretical gaps motivate us to rigorously investigate SA-FL. Toward this end, to fully understand the impact of incomplete client participation on conventional FL, we first show that conventional FL is *not* PAC-learnable under incomplete client participation in the worst case. Then, we show that the PAC-learnability of FL with incomplete client participation can indeed be revived by SA-FL, which theoretically justifies the use of SA-FL for the first time. Lastly, to provide practical guidance for SA-FL training under *incomplete client participation*, we propose the SAFARI (server-assisted federated averaging) algorithm that enjoys the same linear convergence speedup guarantees as classic FL with ideal client participation assumptions, offering the first SA-FL algorithm with convergence guarantee. Extensive experiments on different datasets show SAFARI significantly improve the performance under incomplete client participation.

## 1 INTRODUCTION

Since the seminal work by McMahan et al. (2017), federated learning (FL) has emerged as a powerful distributed learning paradigm that enables a large number of clients (e.g., edge devices) to collaboratively train a model under a central server's coordination. However, as FL gaining popularity, it has also become apparent that FL faces a key challenge unseen in traditional distributed learning in data-center settings – system heterogeneity. Generally speaking, system heterogeneity in FL is caused by the massively different computation and communication capabilities at each client (computational power, communication capacity, drop-out rate, etc.). Studies have shown that system heterogeneity can significantly impact client participation in a highly non-trivial fashion and render *incomplete client participation*, which severely degrades the learning performance (Bonawitz et al., 2019; Yang et al., 2021a). For example, it is shown in (Yang et al., 2021a) that more than $30\%$ clients never participate in FL, while only $30\%$ of the clients contribute to $81\%$ of the total computation even if the server uniformly samples the clients. Exacerbating the problem is the fact that clients' status could be unstable and time-varying due to the aforementioned computation/communication constraints.

To mitigate the impact of incomplete client participation, one approach called *server-assisted federated learning* (SA-FL) has been widely adopted in real-world FL systems in recent years (see, e.g., (Zhao et al., 2018; Wang et al., 2021b)). The basic idea of SA-FL is to equip the server with a small auxiliary dataset that approximately mimics the population distribution, so that the distribution deviation induced by incomplete client participation can be corrected. Nonetheless, while SA-FL has empirically demonstrated its considerable efficacy in addressing incomplete client participation problem in practice, there remains *a lack of theoretical understanding* for SA-FL. This motivates us to investigate the efficacy of SA-FL against incomplete client participation for FL in this paper.

Somewhat counterintuitively, to understand SA-FL, one must first fully understand the impact of incomplete client participation on conventional FL. In other words, we need to first answer the following fundamental questions: *"1) What are the impacts of incomplete client participation on conventional FL learning performance?"* Upon answering this question, the next important follow-up question regarding SA-FL is: *"2) What benefits could SA-FL bring and how could we theoretically characterize them?* Also, just knowing the benefits of SA-FL is still not enough to provide guidelines on how to appropriately use server-side data to design training algorithms with convergence guarantees. Therefore, our third fundamental question for SA-FL is: *"3) Is it possible to develop SA-FL training algorithms with provable convergence rates that can match the state-of-the-art rates in conventional FL?"* Indeed, answering these three questions constitutes the rest of this paper, where we address the first two questions through the lens of PAC (probably approximately correct) learning, while resolving the third question by proposing a provably convergent SA-FL algorithm. Our major contributions in this work are summarized as follows:

- By establishing a *worst-case* generalization error lower bound, we show that classic FL is *not* PAC-learnable under incomplete client participation. In other words, no learning algorithm can approach zero generalization error with incomplete client participation for classic FL even in the limit of infinitely many data samples and training iterations. This insight, though being negative, warrants the necessity of developing new algorithmic techniques and system architectures (e.g., SA-FL) to modify the classic FL framework to mitigate incomplete client participation.

- We prove a new generalization error bound to show that SA-FL can indeed *revive the PAC learnability of FL* with incomplete client participation. We note that this bound could reach zero asymptotically as the number data samples increases. This is much stronger than previous results in domain adaptation with non-vanishing small error (see Section 2 for details).

- To ensure that SA-FL is provably convergent in training, we propose a new training algorithm for SA-FL called SAFARI (server-assisted federated averaging). By carefully designing the server-client update coordination, we show that SAFARI achieves an $\mathcal{O}(1/\sqrt{mkR})$ convergence rate to a stationary point, matching the convergence rates of state-of-the-art classic FL algorithms. This shows SAFARI can enjoy the same benefits of parallelism in SA-FL under incomplete client participation, representing a significant practical improvement over existing approaches. We also conduct extensive experiments to demonstrate the efficacy and efficiency of our SAFARI algorithm.

The rest of this paper is organized as follows. In Section 2, we review the literature to put our work in comparative perspectives. Section 3 presents the PAC learning analysis of standard FL under incomplete participation and our proposed SA-FL framework. We then propose the SAFARI algorithm with convergence guarantees in Section 4, followed by extensive experiments in Section 5.

## 2 RELATED WORK

**1) Client Participation in Federated Learning:** The seminal FedAvg algorithm was first proposed in McMahan et al. (2017) as a heuristic to improve communication efficiency and data privacy for FL. Since then, there have been many follow-ups (e.g., (Li et al., 2020a; Wang et al., 2020; Zhang et al., 2020; Acar et al., 2021; Karimireddy et al., 2020; Luo et al., 2021; Mitra et al., 2021; Karimireddy et al., 2021; Khanduri et al., 2021; Murata & Suzuki, 2021; Avdiukhin & Kasiviswanathan, 2021) and so on) on addressing the data heterogeneity challenge in FL. However, most of these works (e.g., (Li et al., 2020a; Wang et al., 2020; Zhang et al., 2020; Acar et al., 2021; Karimireddy et al., 2020; Yang et al., 2021b)) are based on the full or uniform (i.e., sampling clients uniformly at random) client participation assumption. The full or uniform participation assumptions are essential since they are required to ensure that the stochastic gradient estimator is unbiased in each round of update. Thus, even if "model drift" or "objective inconsistency" emerge due to local updates (Karimireddy et al., 2020; Wang et al., 2020), the full/uniform client participation in each communication round averages them out in the long run, thus guaranteeing convergence. A related line of works in FL different from full/uniform client participation focuses on *proactively creating* flexible client participation (see, e.g., (Xie et al., 2019; Ruan et al., 2021; Gu et al., 2021; Avdiukhin & Kasiviswanathan, 2021; Yang et al., 2022; Wang & Ji, 2022)). The main idea here is to allow asynchronous communication or fixed participation pattern (e.g., given probability) for clients to flexibly participate in training. Existing works in this area often require extra assumptions, such as bounded delay (Ruan et al., 2021; Gu et al., 2021; Yang et al., 2022; Avdiukhin & Kasiviswanathan, 2021) and identical computation

rate (Avdiukhin & Kasiviswanathan, 2021). Under these assumptions, although stochastic gradients are no longer unbiased estimators of full gradients, the deviation in each communication round remains bounded. For sufficiently many rounds, the impact of such deviation from full gradients vanishes asymptotically, since each client can *still* participate in FL in the long run. In contrast, this paper considers a more practical worst-case scenario in FL – *incomplete client participation* even in the long run, which can be caused by many heterogeneous factors as mentioned in Section 1.

**2) Domain Adaptation:** Since incomplete client participation induces a gap between the dataset distribution used for FL training and the true data population distribution across all clients, our work is also related to the field of domain adaptation in learning. Domain adaptation focuses on the learnability of a model trained in one source domain but applied to a different and related target domain. The basic approach is to quantify the error in terms of the source domain plus the distance between source and target domains. Specifically, let $P$ and $Q$ be the target and source distributions, respectively. Then, the generalization error is expressed as $\mathcal{O}(\mathcal{A}(n_Q)) + dist(P, Q)$, where $\mathcal{A}(n_Q)$ is an upper bound of the error dependent on the total number of samples in $Q$. Widely-used distance measures include $d_{\mathcal{A}}$-divergence (Ben-David et al., 2010; David et al., 2010) and $\mathcal{Y}$-discrepancy (Mansour et al., 2009; Mohri & Medina, 2012). We note, however, that results in domain adaptation is not directly applicable in FL with incomplete client participation, since doing so yields an overly pessimistic bound. Specifically, the error based on domain adaptation remains non-zero for asymptotically small distance $dist(P, Q)$ between $P$ and $Q$ even with infinite many samples in $n_Q$ (i.e., $\mathcal{A}(n_Q) \to 0$). In this paper, rather than directly using results from domain adaptation, we establish a much *sharper* upper bound (see Section 3). Another line of work in domain adaptation uses importance weights defined by the density ratios between $P$ and $Q$ to correct the bias and reduce the discrepancy (Sugiyama et al., 2007a;b; Cortes et al., 2008). However, due to FL privacy constraints, such density ratios are difficult to estimate, rendering importance-weights-based methods infeasible in FL. A closely related work is (Hanneke & Kpotufe, 2019), which proposed a new notion of discrepancy between source and target distributions called transfer exponents. However, this work considers *non-overlapping* support between $P$ and $Q$, while we focus on *overlapping* support naturally implied by FL systems (see Fig. 3.2 in Section 3.2).

# 3    PAC-LEARNABILITY OF FL WITH INCOMPLETE CLIENT PARTICIPATION

In this section, we first focus on understanding the impacts of incomplete client participation on conventional FL in terms of PAC-learning in Section 3.1. This will also pave the way for studying SA-FL later in Section 3.2. In what follows, we start with the conventional FL formulation and some definitions in statistical learning that are necessary to formulate and prove our main results.

The goal of an $M$-client FL system is to minimize the following loss function $F(\mathbf{x}) = \mathbb{E}_{i \sim \mathcal{P}}[F_i(\mathbf{x})]$, where $F_i(\mathbf{x}) \triangleq \mathbb{E}_{\xi \sim P_i}[f_i(\mathbf{x}, \xi)]$. Here, $\mathcal{P}$ represents the distribution of the entire client population, $\mathbf{x} \in \mathbb{R}^d$ is the model parameter, $F_i(\mathbf{x})$ represents the local loss function at client $i$, and $P_i$ is the underlying distribution of the local dataset at client $i$. In general, due to data heterogeneity, $P_i \neq P_j$ if $i \neq j$. However, the loss function $F(\mathbf{x})$ or full gradient $\nabla F(\mathbf{x})$ can not be directly computed since the exact distribution of data is unknown in general. Instead, one often considers the following empirical risk minimization (ERM) problem in the finite-sum form based on empirical risk $\hat{F}(\mathbf{x})$:

$$\min_{\mathbf{x} \in \mathbb{R}^d} \hat{F}(\mathbf{x}) = \sum_{i \in [M]} \alpha_i \hat{F}_i(\mathbf{x}) \triangleq (1/|S_i|) \sum_{\xi \in S_i} f_i(\mathbf{x}, \xi),$$

where $S_i$ is a local dataset at client $i$ with cardinality $|S_i|$, whose samples are independently and identically sampled from local distribution $P_i$, and $\alpha_i = |S_i|/(\sum_{j \in [M]} |S_j|)$ (hence $\sum_{i \in [M]} \alpha_i = 1$). For simplicity, we consider the balanced dataset case: $\alpha_i = 1/M, \forall i \in [M]$, but we note our results can be straightforwardly generalized to unbalanced dataset settings at the expense of more complex notations. Next, we state several definitions from statistical learning theory (Mohri et al., 2018).

**Definition 1** (Generalization Error and Empirical Error). *Given a hypothesis $h \in \mathcal{H}$, a target concept $f$, an underlying distribution $\mathcal{D}$ and a dataset $S$ i.i.d. sampled from $\mathcal{D}$ ($S \sim \mathcal{D}$), the generalization error and empirical error of $h$ are defined as follows: $\mathcal{R}_{\mathcal{D}}(h, f) = \mathbb{E}_{(x,y) \sim \mathcal{D}} l(h(x), f(x))$ and $\hat{\mathcal{R}}_D(h, f) = (1/|S|) \sum_{i \in S} l(h(x_i), f(x_i))$, where $l(\cdot)$ is some valid loss function.*

For notational simplicity, we use $\mathcal{R}_\mathcal{D}(h)$ and $\hat{\mathcal{R}}_D(h)$ for generalization and empirical errors and omit target concept $f$ whenever it is clear from the context.

**Definition 2** (Optimal Hypothesis). *For a distribution $\mathcal{D}$ and a dataset $S \sim \mathcal{D}$, we define $h_\mathcal{D}^* = \underset{h \in \mathcal{H}}{\operatorname{argmin}} \mathcal{R}_\mathcal{D}(h)$ and $\hat{h}_\mathcal{D}^* = \underset{h \in \mathcal{H}}{\operatorname{argmin}} \hat{\mathcal{R}}_\mathcal{D}(h)$.*

**Definition 3** (Excess Error). *For hypothesis $h$ and distribution $\mathcal{D}$, the excess error and excess empirical error are defined as e respectively.*

### 3.1 CONVENTIONAL FEDERATED LEARNING WITH INCOMPLETE CLIENT PARTICIPATION

With the above notations, we now study conventional FL with incomplete client participation. Consider an FL system with $M$ clients in total. We let $P$ denote the underlying joint distribution of the entire system, which can be decomposed into the summation of the local distributions at each client, i.e., $P = \sum_{i \in [M]} \lambda_i P_i$, where $\lambda_i > 0$ and $\sum_{i \in [M]} \lambda_i = 1$. We assume that each client $i$ has $n$ training samples i.i.d. drawn from $P_i$, i.e., $|S_i| = n, \forall i \in [M]$. Then, $S = \{(x_i, y_i), i \in [M \times n]\}$ can be viewed as the dataset i.i.d. sampled from the joint distribution $P$. We consider an incomplete client participation setting, where $m \in [0, M)$ clients participate in the FL training as a result of some client sampling/participation process $\mathcal{F}$. We let $\mathcal{F}(S)$ represent the data ensemble actually used in training and $\mathcal{D}$ denote the underlying distribution corresponding to $\mathcal{F}(S)$. For convenience, we define the notion $\omega = \frac{m}{M}$ as the *FL system capacity* (i.e., only $m$ clients participate in the training). For FL with incomplete client participation, we establish the following fundamental performance limit of any FL learner in general. For simplicity, we use binary classification with zero-one loss here, but it is already sufficient to establish the PAC learnability lower limit.

**Theorem 1** (Impossibility Theorem). *Let $\mathcal{H}$ be a non-trivial hypothesis space and $\mathcal{L} : (\mathcal{X}, \mathcal{Y})^{(m \times n)} \to \mathcal{H}$ be the learner for an FL system. There exists a client participation process $\mathcal{F}$ with FL system capacity $\omega$, a distribution $P$, and a target concept $f \in \mathcal{H}$ with $\min_{h \in \mathcal{H}} \mathcal{R}_P(h, f) = 0$, such that $\mathbb{P}_{S \sim P}\left[\mathcal{R}_P(\mathcal{L}(\mathcal{F}(S), f)) > \frac{1 - \omega}{8}\right] > \frac{1}{20}$.*

*Proof Sketch.* The proof is based on the method of induced distributions in (Bshouty et al., 2002; Mohri et al., 2018; Konstantinov et al., 2020). We first show that the learnability of an FL system is equivalent to that of a system that arbitrarily selects $mn$ out of $Mn$ samples in the centralized learning. Then, for any learning algorithm, there exists a distribution $P$ such that dataset $\mathcal{F}(S)$ resulting from incomplete participation and seen by the algorithm is always distributed identically for any target functions. Thus, no algorithm can learn a better predictor than random guessing. Due to space limitation, we relegate the full proof to supplementary material. □

Given the system capacity $\omega \in (0, 1)$, the above theorem characterizes the worst-case scenario for FL with incomplete client participation. It says that for any learner (i.e., algorithm) $\mathcal{L}$, there exist a bad client participation process $\mathcal{F}$ and distributions $P_i, i \in [M]$ over target function $f$, for which the error of the hypotheses returned by $\mathcal{L}$ is constant with non-zero probability. In other words, FL with incomplete client participation is *not PAC-learnable*. One interesting observation here is that the lower bound is *independent* of the number of samples per client $n$. This indicates that even if each client has *infinitely many* samples ($n \to \infty$), it is impossible to have a zero-generation-error learner under the incomplete client participation (i.e., $\omega \in (0, 1)$). Note that this fundamental result relies on two conditions: *heterogeneous* dataset and *arbitrary* client participation. Under these two conditions, there exists a worst-case scenario where the underlying distribution $\mathcal{D}$ of the participating data $S_\mathcal{D} = \mathcal{F}(S)$ deviates from the ground truth $P$, thus yielding a non-vanishing error.

This result sheds light on system and algorithm design for FL. That is, how to motivate client participation in FL effectively and efficiently: the participating client's data should be comprehensive enough to model the complexity of the joint distribution $P$ to close the gap between $\mathcal{D}$ and $P$. Note that this result is not contradictory to previous works where the convergence of FedAvg is guaranteed, since this theorem is not applicable for homogeneous (i.i.d.) datasets or uniformly random client participation. As mentioned earlier, most of the existing works rely on at least one of these two assumptions. However, none of these two assumptions hold for conventional FL with incomplete client participation in practice. In addition to system heterogeneity, other factors such as Byzantine attackers could also render incomplete client participation. For example, even for full client participation in FL, if part of the clients are Byzantine attackers, the impossibility theorem also

applies. Thus, our impossibility theorem also justifies the empirical use of server-assisted federated learning (i.e., FL with server-side auxiliary data) to build trust (Cao et al., 2021).

## 3.2 THE PAC-LEARNABILITY OF SERVER-ASSISTED FEDERATED LEARNING (SA-FL)

The intuition of SA-FL is to utilize a dataset $T$ i.i.d. sampled from distribution $P$ with cardinality $|T| = n_T$ as a vehicle to correct potential distribution deviations due to incomplete client participation. By doing so, the server steers the learning by a small number of representative data, while the clients assist the learning by federation to leverage the huge amount of privately decentralized data ($n_S \gg n_T$). Note that the assumption of having a server-side dataset is not restrictive since such datasets are already available in many FL systems: although not always necessary for training, an auxiliary dataset is often needed for defining FL tasks (e.g., simulation prototyping) before training and model checking after training (e.g., quality evaluation and sanity checking) (McMahan et al., 2021; Wang et al., 2021a). Also, obtaining an auxiliary dataset is affordable since the number of data points required is relatively small, and hence the cost is low. Then, SA-FL can be easily achieved or even with manually labelled data thanks to its small size. It is also worth noting that many works use such auxiliary datasets in FL for security (Cao et al., 2021), incentive design (Wang et al., 2019), and knowledge distillation (Cho et al., 2021).

For SA-FL, we consider the same incomplete client participation setting that induces a dataset $S_{\mathcal{D}} \sim \mathcal{D}$ with cardinality $n_S$ and $\mathcal{D} \neq P$. As a result, the learning process is to minimize $\mathcal{R}_P(h)$ by utilizing $(\mathcal{X}, \mathcal{Y})^{n_T + n_S}$ to learn a hypothesis $h \in \mathcal{H}$. For notional clarity, we assume the joint dataset $S_Q = (S_{\mathcal{D}} \cup T) \sim Q$ with cardinality $n_T + n_S$ for some distribution $Q$. Before deriving the generalization error bound for SA-FL, we state the following assumption and definition.

**Assumption 1** (Noise Condition). *Suppose $h_P^*$ and $h_Q^*$ exist. There exist $\beta_P, \beta_Q \in [0, 1]$ and $\alpha_P, \alpha_Q > 0$ s.t., $\mathbb{P}_{x \sim P}(h(x) \neq h_P^*(x)) \leq \alpha_P[\varepsilon_P(h)]^{\beta_P}$, $\mathbb{P}_{x \sim Q}(h(x) \neq h_Q^*(x)) \leq \alpha_q[\varepsilon_Q(h)]^{\beta_Q}$.*

This assumption is a traditional noise model known as the Bernstein class condition, which has been widely used in the literature (Massart & Nédélec, 2006; Koltchinskii, 2006; Hanneke, 2016).

**Assumption 2** (($\alpha, \beta$)-Positively-Related). *Distributions $P$ and $Q$ are said to be ($\alpha, \beta$)-positively-related if there exist constants $\alpha \geq 0$ and $\beta \geq 0$ such that $|\varepsilon_P(h) - \varepsilon_Q(h)| \leq \alpha[\varepsilon_Q(h)]^\beta, \forall h \in \mathcal{H}$.*

Assumption 2 specifies a stronger constraint between distributions $P$ and $Q$. It implies that the difference of excess error for one hypothesis $h \in \mathcal{H}$ between $P$ and $Q$ is bounded by the excess error of $Q$ in some exponential form. Assumption 2 is one of the major *novelty* in our paper and unseen in the literature. We note that this ($\alpha, \beta$)-positively-related condition is a mild condition. To see this, consider the following "one-dimensional" example for simplicity. Let $\mathcal{H}$ be the class of hypotheses defined on the real line: $\{h_t = t, t \in R\}$, and let two uniform distributions be $P := \mathcal{U}[a, b]$ and $Q := \mathcal{U}[a', b']$. Due to the incomplete client sampling in FL, the support of $Q$ is a subset of that of $P$, i.e., $a \leq a' \leq b' \leq b$. Denote the target hypothesis $t^* \in [a', b']$. Then, for any hypothesis $h_t$ with threshold $t$, we have $\epsilon_P(h_t) = \frac{|t - t^*|}{b - a}$ and $\epsilon_Q(h_t) = \frac{|t - t^*|}{b' - a'}$. That is, our "($\alpha, \beta$)-Positively-Related" holds for $\alpha = 1 - \frac{b' - a'}{b - a}$ and $\beta = 1$. The above "one-dimensional" example can be further extended to general high-dimensional cases as follows. Intuitively, the difference of excess errors of $P$ and $Q$ (i.e., $|\epsilon_P(h) - \epsilon_Q(h)|$) is a function in the form of $\int_S |Q_X - P_X| dS$ for a common support domain $S \subset supp(Q)$. Thus, the "($\alpha, \beta$)-Positively-Related" condition can be written as $|\int_S Q_X dS - \int_S P_X dS| \leq \alpha(\int_S Q_X)^\beta$. If distribution $Q$ has more probability mass over $S$ than distribution $P$, choosing $\beta = 1$ and $\alpha$ to be a sufficiently large constant clearly satisfies the ($\alpha, \beta$)-positively-related condition. Otherwise, letting $\beta \to 0$ and choosing $\alpha$ to be a sufficiently large constant satisfies the ($\alpha, \beta$)-positively-related condition with probability one.

With the above assumption and definition, we have the following generation error bound for SA-FL, which shows that SA-FL is PAC-learnable:

**Theorem 2** (Generalization Error Bound for SA-FL). *For an SA-FL system with arbitrary system and data heterogeneity, if distributions $P$ and $Q$ satisfy Assumption 1 and are ($\alpha, \beta$)-positively-related, then with probability at least $1 - \delta$ for any $\delta \in (0, 1)$, it holds that*

$$\varepsilon_P(\hat{h}_Q^*) = \widetilde{\mathcal{O}}\left( \left( \frac{d_{\mathcal{H}}}{n_T + n_S} \right)^{\frac{1}{2 - \beta_Q}} + \left( \frac{d_{\mathcal{H}}}{n_T + n_S} \right)^{\frac{\beta}{2 - \beta_Q}} \right), \tag{1}$$

*where $d_{\mathcal{H}}$ denotes the finite VC dimension for hypotheses class $\mathcal{H}$.*

Note the generalization error bound of centralized learning is $\widetilde{\mathcal{O}}((\frac{d_{\mathcal{H}}}{n})^{\frac{1}{2-\beta_Q}})$ (hiding logarithmic factors) with $n$ samples in total and noise parameter $\beta_Q$ (Hanneke, 2016). Note that when $\beta \geq 1$, the first term in Eq. (1) dominates. Hence, Theorem 2 implies that the generalization error bound in this case for SA-FL *matches* that of centralized learning (with dataset size $n_T + n_S$). Meanwhile, for $0 < \beta < 1$, compared with solely training on server's dataset $T$, SA-FL exhibits an improvement from $\widetilde{\mathcal{O}}((\frac{1}{n_T})^{\frac{1}{2-\beta_Q}})$ to $\widetilde{\mathcal{O}}((\frac{1}{n_T+n_S})^{\frac{\beta}{2-\beta_Q}})$.

Note that SA-FL shares some similarity with the domain adaptation problem, where the learning is on $Q$ but the results will be adapted to $P$. In what follows, we offer some deeper insights between the two by answering two key questions: *1) What is the difference between SA-FL and domain adaptation (a.k.a. transfer learning)?* and *2) Why is SA-FL from $Q$ to $P$ PAC-learnable, but FL from $D$ to $P$ with incomplete client participation not PAC-learnable (as indicated in Theorem 1)?*

To answer these questions, we illustrate the distribution relationships for domain adaptation and federated learning, in Fig.1, respectively. In domain adaptation, the target $P$ and source $Q$ distributions often have overlapping support but there also exists *distinguishable difference*. In contrast, the two distributions $P$ and $Q$ in SA-FL happen to share exactly the *same support* with different density, since $Q$ is a *mixture* of $D$ and $P$. As a result, the known bounds in domain adaptation (or transfer learning) are pessimistic for SA-FL. For example, the $dist(P, Q)$ in $d_{\mathcal{A}}$-divergence and $\mathcal{Y}$-divergence both have non-negligible gaps when applied to SA-FL. Here in Theorem 2, we provide a generalization error bound in terms of the total sample size $n_T + n_S$, thus showing the benefit of SA-FL.

Moreover, for SA-FL, only the auxiliary dataset $T \overset{i.i.d.}{\sim} P$ is directly available to the server. The clients' datasets could be used in SA-FL training, but they are not directly accessible due to privacy constraints. Thus, previous methods in domain adaptation (e.g., importance weights-based methods in covariate shift adaptation (Sugiyama et al., 2007a;b)) are *not* applicable since they require the knowledge of density ratio between training and test datasets.

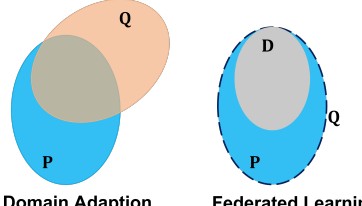

Figure 1: Diagram of distribution supports for domain adaptation and federated learning.

The key difference between FL and SA-FL lies in relations among $D, P$ and $Q$. For FL, the distance between $D$ and $P$ with incomplete participation could be large due to system and data heterogeneity in the worst-case. More specifically, the support of $D$ could be narrow enough to miss some part of $P$, resulting in non-vanishing error as indicated in Theorem 1. For SA-FL, distribution $Q$ is a mixture of $P$ and $D$ ($Q = \lambda_1 D + \lambda_2 P$, with $\lambda_1, \lambda_2 \geq 0$, $\lambda_1 + \lambda_2 = 1$), thus having the same support with $P$. Hence, under Assumption 2, the PAC-learnability is guaranteed. Although we provide a promising bound to show the PAC-learnability of SA-FL in Theorem 2, the superiority of SA-FL over training solely with dataset $T$ in server (i.e., $\widetilde{\mathcal{O}}((\frac{1}{n_T})^{\frac{1}{2-\beta_P}})$) is not always guaranteed as $\beta \to 0$ (i.e., $Q$ becomes increasingly different from $P$). In what follows, we reveal under what conditions could SA-FL perform *no worse than* centralized learning.

**Theorem 3** (Conditions of SA-FL Being No Worse Than Centralized Learning). *Consider an SA-FL system with arbitrary system and data heterogeneity. If Assumption 1 holds and additionally $\hat{\mathcal{R}}_P(\hat{h}_Q^*) \leq \hat{\mathcal{R}}_P(h_Q^*)$ and $\varepsilon_P(h_Q^*) = \mathcal{O}(\mathcal{A}(n_T, \delta))$, where $\mathcal{A}(n_T, \delta) = \frac{d_{\mathcal{H}}}{n_T} \log(\frac{n_T}{d_{\mathcal{H}}} + \frac{1}{n_T} \log(\frac{1}{\delta}))$, then with probability at least $1 - \delta$ for any $\delta \in (0, 1)$, it holds that $\varepsilon_P(\hat{h}_Q^*) = \widetilde{\mathcal{O}}\left((d_{\mathcal{H}}/n_T)^{\frac{1}{2-\beta_P}}\right)$.*

Here, we remark that $\varepsilon_P(h_Q^*) = \mathcal{O}(\mathcal{A}(n_T, \delta))$ is a weaker condition than the $\varepsilon_P(h_Q^*) = 0$ condition and the covariate shift assumption ($P_{Y|X} = Q_{Y|X}$) used in the transfer learning literatures (Hanneke & Kpotufe, 2019; 2020). Together with the condition $\hat{\mathcal{R}}_P(\hat{h}_Q^*) \leq \hat{\mathcal{R}}_P(h_Q^*)$, the following intermediate result holds: $\hat{\mathcal{R}}_P(\hat{h}_Q^*) - \hat{\mathcal{R}}_P(h_P^*) = \mathcal{O}(A(n_T, \delta))$ (see Lemma 2 in the supplementary material). Intuitively, this states that "if $P$ and $Q$ share enough similarity, then the difference of excess empirical error between $\hat{h}_Q^*$ and $h_P^*$ on $P$ can be bounded." Thus, the excess error of $\hat{h}_Q^*$ shares the same upper bound as that of $\hat{h}_P^*$ in centralized learning. Therefore, Theorem 3 implies that, under mild conditions,

---

**Algorithm 1** The SAFARI Algorithm for SA-FL.

---

1: Initialize model $\mathbf{x}_0$, iteration index $t = 0$.
2: **for** $r = 0, \cdots, R - 1$ **do**
3:     With probability $q$:                               ★ client update round $r \in \mathcal{T}_c$
4:         The server samples clients $S_r$ and send current model $\mathbf{x}_r$.
5:         Each client $i \in S_r$ computes in parallel:
6:             Local update: $\mathbf{x}_{r,k+1}^i = \mathbf{x}_{r,k}^i - \eta_c \nabla F_i(\mathbf{x}_{r,k}^i, \xi_{r,k}^i), k \in [K]$ starting from $\mathbf{x}_{r,0}^i = \mathbf{x}_t$.
7:             Send $\mathbf{x}_r^i = \mathbf{x}_{r,K+1}^i$ to server.
8:         Server aggregation: $\mathbf{x}_{r+1} = \frac{1}{|S_r|} \sum_{i \in S_r} \mathbf{x}_r^i$.
9:     Otherwise                                     ★ server update round $r \in \mathcal{T}_s$
10:         Server update: $\mathbf{x}_{r+1} = \mathbf{x}_r - \eta_s \nabla F(\mathbf{x}_r, \xi_r)$.
11: **end for**

---

SA-FL guarantees the same generalization error upper bound as that of centralized learning, hence being "no worse than" centralized learning with dataset $T$.

Last but not least, it is worth pointing out that, for ease of illustration, Theorem 2–3 are based on the assumption that the auxiliary dataset $T \overset{i.i.d.}{\sim} P$. Nonetheless, it is of practical importance to consider the scenario where $T$ is sampled from a related but slightly different distribution $P'$ rather than the target distribution $P$ itself. In fact, the above assumption could be relaxed to $T \overset{i.i.d.}{\sim} P'$ for any $P'$ as long as the mixture distribution $Q = \lambda_1 D + \lambda_2 P'$ is $(\alpha, \beta)$-positively-related with $P$. Under such condition, we can show that the main results in Theorem 2–3 continue to hold.

## 4 THE SAFARI ALGORITHM FOR TRAINING UNDER SA-FL

In Section 3, we have shown that SA-FL is PAC-learnable with incomplete client participation. In this section, we turn our attention to the *training* of the SA-FL regime with incomplete client participation, which is also under-explored in the literature. First, we note that the standard FedAvg algorithm may fail to converge to a stationary point with incomplete client participation as indicated by previous works (Yang et al., 2022). Now with SA-FL, we aim to answer the following questions:

*1) Under SA-FL, how should we appropriately use the server-side dataset to develop training algorithms in the SA-FL regime to achieve provable stationary point convergence guarantee?*

*2) If Question 1) can be resolved, could we further achieve the same convergence rate in SA-FL training with incomplete client participation as that in traditional FL with ideal client participation?*

In this section, we resolve the above questions affirmatively by proposing a new algorithm called SAFARI (server-assisted federated averaging) for SA-FL with theoretically provable convergence guarantees. As shown in Algorithm 1, SAFARI contains two options in each round, client update option or global server update option. For a communication round $r \in \{0, \cdots, R - 1\}$, with probability $q \in [0, 1]$, the client update option is chosen (i.e., $r \in \mathcal{T}_c$), where local updates are executed by clients in the current participating client set $S_r$ in a similar fashion as the FedAvg (McMahan et al., 2017). Specifically, the client update option performs the following three steps: 1) Server samples a subset of clients $S_r$ as in conventional FL and synchronizes the latest global model $\mathbf{x}_r$ with each participating clients in $S_r$ (Line 4); 2) All participating clients initialize their local models as $\mathbf{x}_r$ and then perform $K$ local steps following the stochastic gradient descent (SGD) method. Then, each participating client sends its locally updated model $\mathbf{x}_r^i = \mathbf{x}_{r,K+1}^i$ back to the server (Lines 5-7); 3) Upon receiving the local update $\mathbf{x}_r^i$, the server aggregates and updates the global model (Line 8). On the other hand, with probability $1 - q$, the server update option is chosen (i.e,. $r \in \mathcal{T}_s$), where the server updates the global model with its auxiliary data following the SGD (Line 10).

We note that SAFARI can be viewed as a mixture of the FedAvg algorithm with client-side datasets (cf. the client update option) and a centralized SGD algorithm using the server-side dataset only (cf. the server update option), which are governed by a probability parameter $q$. The basic idea of this two-option approach is to leverage client-side parallel computing to accelerate the training process, while using the server-side dataset to mitigate the bias caused by incomplete client participation. We will show later that, by appropriately choosing the $q$-value, SAFARI simultaneously achieves the

stationary point convergence and linear convergence speedup. Before presenting the convergence performance results, we first state three commonly used assumptions in FL.

**Assumption 3.** *(L-Lipschitz Continuous Gradient) There exists a constant $L > 0$, such that $\|\nabla F(\mathbf{x}) - \nabla F(\mathbf{y})\| \leq L\|\mathbf{x} - \mathbf{y}\|, \forall \mathbf{x}, \mathbf{y} \in \mathbb{R}^d$.*

**Assumption 4.** *The stochastic gradient calculated by the client or server is unbiased with bounded variance: $\mathbb{E}[\nabla f(\mathbf{x}, \xi)] = \nabla f(\mathbf{x})$, and $\mathbb{E}[\|\nabla f(\mathbf{x}, \xi) - \nabla f(\mathbf{x})\|^2] \leq \sigma^2$.*

**Assumption 5.** *(Bounded Gradient Dissimilarity) $\|\nabla F_i(\mathbf{x}) - \nabla F(\mathbf{x})\|^2 \leq \sigma_G^2, \forall i \in [M]$.*

With the assumptions above, we state the main convergence result of SAFARI as follows:

**Theorem 4** (Convergence Rate for SAFARI ). *Under Assumptions 3 - 5, if $\eta_c \leq \frac{1}{4\sqrt{30}LK}$, $\eta_c = \frac{2\eta_s}{K}$, and $q \leq 1 / \left( \frac{4\sigma_G^2 - 4G_2(\frac{1}{2K^2} - \frac{2L\eta_s^2}{K^2})}{(1-L\eta_s)G_1} + 1 \right)$, then, the sequence $\{\mathbf{x}_r\}$ generated by SAFARI satisfies:*

$$\frac{1}{R}\sum_{r=1}^{R}\mathbb{E}\|\nabla F(\mathbf{x}_r)\|^2 \leq \frac{2(F(\mathbf{x}_0) - F(\mathbf{x}^*))}{R\eta_s} + L\eta_s(1-q)\sigma^2 + \frac{80qL^2\eta_s^2}{K}(\sigma^2 + 6K\sigma_G^2) + \frac{8Lq\eta_s}{mK}\sigma^2,$$

*where $G_1 = \min_{r \in \mathcal{T}_s} \|\nabla F(\mathbf{x}_r)\|^2$ and $G_2 = \min_{r \in \mathcal{T}_c} \left\| \frac{1}{m}\sum_{i\in[m]}\sum_{k\in[K]}\nabla F_i(\mathbf{x}_{r,k}^i) \right\|^2$.*

Theorem 4 says that, by using the server-side update with an appropriately chosen $q$-value, SAFARI effectively mitigates the bias that arises from incomplete client participation. With proper probability $q$, SAFARI guarantees the stationary point convergence.

Also, it is insightful to point out that SAFARI is a *unifying* framework that includes two classic algorithms as special cases under two extreme settings: i) the i.i.d. client-side data case and the ii) the heterogeneous client-side data case with *unbounded* gradient dissimilarity. In the i.i.d. case, the client-side data are homogeneous, i.e., $F_i(\mathbf{x}) = F$ and $\sigma_G = 0$. In this ideal setting, we can simply choose $q = 1$ and SAFARI degenerates to the classic FedAvg algorithm. In the heterogeneous case with unbounded gradient dissimilarity (i.e., $\sigma_G = \infty$), we can set $q = 0$ (i.e., $|\mathcal{T}_c| = 0$) such that SAFARI degenerates to centralized SGD algorithm. In this heterogeneous setting, Theorem 4 also recover the classic SGD bound by cancelling the $\sigma_G$-dependent terms in the bound.

Further, Theorem 4 immediately implies that, by choosing parameters $q$ and the learning rate $\eta$ appropriately, we achieve linear convergence speedup to a stationary point:

**Corollary 1.** *If $\eta_s = \frac{\sqrt{mK}}{\sqrt{R}}$ and $q = \Omega(1 - \frac{1}{mK})$, SAFARI achieves an $\mathcal{O}(\frac{1}{\sqrt{mKR}})$ convergence rate to a stationary point, implying a linear convergence speedup.*

Corollary 1 suggests that, thanks to the two "control knobs" $\eta_s$ and $q$ in SAFARI , under mild conditions (see further discussions in Appendix, we can avoid FedAvg's limitation that it can only converge to an error ball dependent on the data heterogeneity parameter $\sigma_G$ (Yang et al., 2022). Furthermore, SAFARI with *incomplete client participation* still achieves the *same* convergence rate as that of classic FL algorithms with *ideal client participation*.

## 5 NUMERICAL RESULTS

In this section, we conduct numerical experiments to verify our theoretical results using 1) logistic regression (LR) on MNIST dataset (LeCun et al., 1998) and 2) convolutional neural network (CNN) on CIFAR-10 dataset (Krizhevsky et al., 2009). To simulate data heterogeneity, we distribute the data into each client evenly in a label-based partition, following the same process as in previous works (McMahan et al., 2017; Yang et al., 2021b; Li et al., 2020b). As a result, we can use a parameter $p$ to represent the classes of labels in each client's dataset, which serves as an index of data heterogeneity level (non-i.i.d. index). The smaller $p$-value, the more heterogeneous the data among clients. To mimic incomplete client participation, we force $s$ clients to be excluded. We can use $s$ as an index to represent the degree of incomplete client participation. In our experiments, there are $M = 10$ clients in total, and $m = 5$ clients participate in the training in each communication round, who are uniformly sampled from the $M - s$ clients. We use FedAvg without any server-side dataset as the baseline to compare with SAFARI. Due to space limitation, we highlight four key observations in this section, and relegate all other experimental details and results to the supplementary material.

**1) Performance Degradation of Incomplete Client Participation:** We first show the test accuracy of FedAvg on MNIST for different values of non-i.i.d. index $p$ and incomplete client index $s$ in Fig. 2. For nearly homogeneous data (e.g., $p = 10, 5$), incomplete client participation has negligible impacts on test accuracy. However, for highly non-i.i.d. cases, incomplete client participation results in dramatic performance degradation. Specifically, for $p = 1$, the test accuracy for $s = 4$ is only 57%, yielding a large degradation (35%) compared to that of $s = 0$. This is consistent with the worst-case analysis in Theorem 1 and also the main motivation of SA-FL.

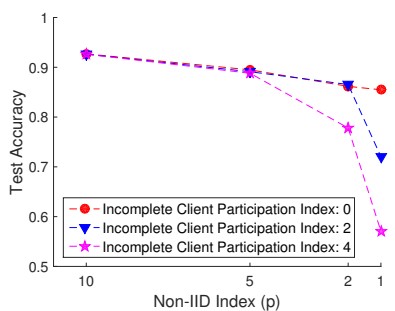

Figure 2: Test accuracy of FedAvg on MNIST.

**2) Improvement of the** SAFARI **Algorithm under Incomplete Client Participation:** In Table 1, we show the test accuracy improvement of our SAFARI algorithm compared with that of FedAvg in standard FL. The key observation is that, with a *small amount* of auxiliary data at the server, there is a significant increase of test accuracy for our SAFARI algorithm. For example, with only 50 data samples at the server ($0.1\%$ of the total training data), there is a $12.75\%$ test accuracy increase. With 1000 data samples, the

Table 1: Test accuracy improvement (%) for SAFARI compared with FedAvg on MNIST with incomplete client participation $s = 4$. '-' means "no statistical difference within 2% error bar".

| SERVER DATASIZE | NON-IID INDEX (P) | | | |
|---|---|---|---|---|
| | 10 | 5 | 2 | 1 |
| 50 | - | - | 4.15 | 12.75 |
| 100 | - | - | 6.55 | 22.19 |
| 500 | - | - | 10.29 | 29.12 |
| 1000 | - | - | 10.88 | 31.42 |

improvement reaches $31.42\%$. This verifies the effectiveness of our SA-FL framework and our SAFARI algorithm. Another observation is that for nearly homogeneous case (e.g., from $p = 10$ to $p = 5$), there is no statistical difference with or without auxiliary data at the server (denoted by '-' in Table 1. This is consistent with the previous observations of negligible degradation in cases with homogeneous data across clients.

**3) saConvergence Speedup of** SAFARI **with Larger Server Dataset:** In this experiment, we illustrate the speedup effect of SAFARI numerically as the size of server dataset increases. In Fig. 3, we show the convergence processes of SAFARI on CIFAR-10 for incomplete client participation ($s = 2$) and non-i.i.d. data ($p = 1$). We can see clearly that the convergence of SAFARI is accelerating and the test accuracy increases as more data are employed at the server. In this experiment setting, we also plot the convergence of FedAvg in Fig. 3. It can be seen that all three cases of SAFARI converge faster than FedAvg in this experiment.

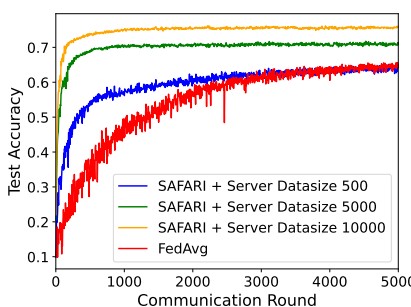

Figure 3: Comparison of test accuracy on CIFAR-10 ($s = 2$, $p = 1$).

## 6 CONCLUSION

In this paper, we rigorously investigated the server-assisted federated learning (SA-FL) framework (i.e., to deploy an auxiliary dataset at the server), which has been increasingly adopted in practice to mitigate the impacts of incomplete client participation in conventional FL. To characterize the benefits of SA-FL, we first showed that conventional FL is *not* PAC-learnable under incomplete client participation by establishing a fundamental generalization error lower bound. Then, we showed that SA-FL is able to revive the PAC-learnability of conventional FL under incomplete client participation. Upon resolving the PAC-learnability challenge, we proposed a new SAFARI (server-assisted federated averaging) algorithm that enjoys convergence guarantee and the same level of communication efficiency as that of conventional FL. Extensive numerical results also validated our theoretical findings.

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
