.* Denote $S$ the dataset with size $Mn$ i.i.d. sampled from distribution $P$, $\mathcal{F}(\cdot)$ the sampling process of FL system, and $\bar{S} = \mathcal{F}(S)$ the training dataset selected by FL system with size $mn$. Consider a distribution $P$ with support on only two points $\{x_1, x_2\}$ such that $\mathbb{P}_P(x_1) = 1 - 4\epsilon$ and $\mathbb{P}_P(x_2) = 4\epsilon$ with $\epsilon = \frac{1 - \omega}{8}$.

First we show that the rare points $x_2$ appears at most $(1 - \omega)Mn$ times with constant probability. Let $\hat{s}$ be the number of $x_2$ points in $S$, then $\hat{s} \sim \mathbb{B}(Mn, \epsilon)$ is a binomial random variable. By the Chernoff bound,

$$\mathbb{P}[\hat{s} \geq (1 - \omega)Mn] = \mathbb{P}[\hat{s} \geq (1 + 1)4\epsilon Mn] \leq e^{-\frac{4\epsilon Mn}{3}} = e^{-\frac{(1 - \omega)Mn}{6}} \leq e^{-\frac{1}{6}} \leq \frac{17}{20}.$$

So $\mathbb{P}[\hat{s} < (1 - \omega)Mn] > \frac{3}{20}$.

Next, we consider the following sampling process with dataset $S = \{(x_1', f(x_1')), \ldots, (x_{M \times n}', f(x_{M \times n}'))\}$: choosing as many data $(x_i', f(x_i')), i \in [mn]$ such that $x_i' = x_1$ as possible to form the training set $\bar{S}$. Let $f_1, f_2 \in \mathcal{H}$ be two target functions whose existence is guaranteed by the non-trivial definition of $\mathcal{H}$ and $f_1(x_1) = f_2(x_1), f_1(x_2) = -f_2(x_2)$, and $\mathcal{S}$ be the set of all datasets in $(\mathcal{X}, \mathcal{Y})^{(M \times n)}$ such that $\hat{s} < (1 - \omega)MN$.

Let $\mathcal{R}(h_s, f) = \mathbb{P}_P[\mathcal{L}(\mathcal{F}(S))(x) \neq f_1(x) \cap x \neq x_1]$, the following holds for these two target functions $f_1$ and $f_2$:

$$\mathcal{R}(h_s, f_1) + \mathcal{R}(h_s, f_2) = \mathbb{P}_P[\mathcal{L}(\mathcal{F}(S))(x) \neq f_1(x) \cap x \neq x_1] + \mathbb{P}_P[\mathcal{L}(\mathcal{F}(S))(x) \neq f_2(x) \cap x \neq x_1]$$
$$= \mathbf{1}_{\mathcal{L}(\mathcal{F}(S))(x_1) \neq f_1(x_1)} \mathbb{P}(x_2) + \mathbf{1}_{\mathcal{L}(\mathcal{F}(S))(x_1) \neq f_2(x_2)} \mathbb{P}(x_1)$$
$$= 4\epsilon.$$

The above result hold in expectation since it holds for any $S \in \mathcal{S}$. Hence, there exists a target function $f \in \mathcal{H}$ such that $\mathbb{E}_{S \in \mathcal{S}} \mathcal{R}(h_s, f) \geq 2\epsilon$. Note $\mathcal{R}(h_s, f) \leq \mathbb{P}(x \neq x_1) = 4\epsilon$, then by decomposing the expectation into two parts we obtain:

$$2\epsilon \leq \mathbb{E}_{S \in \mathcal{S}} \mathcal{R}(h_s, f) = \sum_{S : \mathcal{R}(h_s, f) \geq \epsilon} \mathcal{R}(h_s, f) \mathbb{P}[\mathcal{R}(h_s, f)] + \sum_{S : \mathcal{R}(h_s, f) < \epsilon} \mathcal{R}(\mathcal{R}(h_s, f) \mathbb{P}[\mathcal{R}(h_s, f)]$$
$$\leq 4\epsilon \mathbb{P}_{S \in \mathcal{S}}[\mathcal{R}(h_s, f) \geq 4\epsilon] + \epsilon(1 - \mathbb{P}_{S \in \mathcal{S}}[\mathcal{R}(h_s, f) \geq \epsilon])$$
$$= \epsilon + 3\epsilon \mathbb{P}_{S \in \mathcal{S}}[\mathcal{R}(h_s, f) \geq \epsilon].$$

That is,

$$\mathbb{P}_{S \in \mathcal{S}}[\mathcal{R}(h_s, f) \geq \epsilon] \geq \frac{1}{3}.$$

Note $\mathcal{R}(h_s, f) = \mathbb{P}_P[\mathcal{L}(\mathcal{F}(S))(x) \neq f_1(x) \cap x \neq x_1] \leq \mathcal{R}(\mathcal{L}(\mathcal{F}(S))) = \mathbb{P}_P[\mathcal{L}(\mathcal{F}(S))(x) \neq f_1(x)]$, then we have the final results:

$$\mathbb{P}_{S \sim P}[\mathcal{R}_P(\mathcal{L}(\mathcal{F}(S)), f) \geq \epsilon] \geq \mathbb{P}_{S \sim P}[\mathcal{R}(h_s, f) \geq \epsilon]$$
$$\geq \mathbb{P}_{S \in \mathcal{S}}[\mathcal{R}(h_s, f) \geq \epsilon] \mathbb{P}[S \in \mathcal{S}]$$
$$> \frac{1}{3} \frac{3}{20} = \frac{1}{20}.$$

$\square$

**Theorem 2** (Generalization Error Bound for SA-FL). *For an SA-FL system with arbitrary system and data heterogeneity, if distributions $P$ and $Q$ satisfy Assumption 1 and are $(\alpha, \beta)$-positively-related, then with probability at least $1 - \delta$ for any $\delta \in (0, 1)$, it holds that*

$$\varepsilon_P(\hat{h}_Q^*) = \widetilde{\mathcal{O}}\left(\left(\frac{d_{\mathcal{H}}}{n_T + n_S}\right)^{\frac{1}{2 - \beta_Q}} + \left(\frac{d_{\mathcal{H}}}{n_T + n_S}\right)^{\frac{\beta}{2 - \beta_Q}}\right), \quad (1)$$

*where $d_{\mathcal{H}}$ denotes the finite VC dimension for hypotheses class $\mathcal{H}$.*

*Proof.*

$$\varepsilon_P(\hat{h}_Q^*) = \mathcal{R}_P(\hat{h}_Q^*) - \mathcal{R}_P(h_P^*)$$
$$= [\mathcal{R}_P(\hat{h}_Q^*) - \mathcal{R}_P(h_P^*) - (\mathcal{R}_Q(\hat{h}_Q^*) - \mathcal{R}_Q(h_Q^*))] + \mathcal{R}_Q(\hat{h}_Q^*) - \mathcal{R}_Q(h_Q^*)$$
$$\leq |\varepsilon_P(\hat{h}_Q^*) - \varepsilon_Q(\hat{h}_Q^*)| + \varepsilon_Q(\hat{h}_Q^*)$$
$$\leq \alpha \varepsilon_Q(\hat{h}_Q^*)^\beta + \varepsilon_Q(\hat{h}_Q^*).$$

Combining with Lemma 1, the proof is complete.

**Lemma 1** (Auxiliary Lemma (Massart & Nédélec, 2006; Koltchinskii, 2006; Hanneke & Kpotufe, 2019; 2020)). *For any $m \in \mathbb{N}$ and $\delta \in (0,1)$, define $A(m,\delta) = \frac{d_{\mathcal{H}}}{m} \log(\frac{m}{d_{\mathcal{H}}} + \frac{1}{m} \log(\frac{1}{\delta}))$ With probability at least $1 - \delta$, $\forall h, \hat{h} \in \mathcal{H}$,*

$$\mathcal{R}(h) - \mathcal{R}(\hat{h}) \leq \hat{\mathcal{R}}(h) - \hat{\mathcal{R}}(\hat{h}) + c\sqrt{\min\{\mathbb{P}_S(h \neq \hat{h}), \hat{\mathbb{P}}_S(h \neq \hat{h})\}A(m,\delta)} + cA(m,\delta),$$

$$\frac{1}{2}\mathbb{P}_S(h \neq \hat{h}) - cA(m,\delta) \leq \hat{\mathbb{P}}_S(h \neq \hat{h}) \leq 2\mathbb{P}_S(h \neq \hat{h}) + cA(m,\delta),$$

$$\varepsilon_Q(\hat{h}_Q^*) = [A(m,\delta)]^{\frac{1}{2-\beta_Q}},$$

*where $\mathbb{P}_S(\cdot) = \mathbb{E}[\hat{\mathbb{P}}_S(\cdot)]$, $S$ is the i.i.d. dataset with size $m$ drawn form distribution $Q$, $c \in (0,\infty)$ is a constant.*

$\square$

**Theorem 3** (Conditions of SA-FL Being No Worse Than Centralized Learning). *Consider an SA-FL system with arbitrary system and data heterogeneity. If Assumption 1 holds and additionally $\hat{\mathcal{R}}_P(\hat{h}_Q^*) \leq \hat{\mathcal{R}}_P(h_Q^*)$ and $\varepsilon_P(h_Q^*) = \mathcal{O}(\mathcal{A}(n_T, \delta))$, where $\mathcal{A}(n_T, \delta) = \frac{d_{\mathcal{H}}}{n_T} \log(\frac{n_T}{d_{\mathcal{H}}} + \frac{1}{n_T} \log(\frac{1}{\delta}))$, then with probability at least $1 - \delta$ for any $\delta \in (0,1)$, it holds that $\varepsilon_P(\hat{h}_Q^*) = \widetilde{\mathcal{O}}\left((d_{\mathcal{H}}/n_T)^{\frac{1}{2-\beta_P}}\right)$.*

*Proof.* Without loss of generality, we use $c$ serve as a generic constant since we focus on the order in terms of the sample number and thus omit the constant factor.

$$\varepsilon_P(\hat{h}_Q^*) = \mathcal{R}_P(\hat{h}_Q^*) - \mathcal{R}_P(h_P^*)$$
$$\leq \hat{\mathcal{R}}_P(\hat{h}_Q^*) - \hat{\mathcal{R}}_P(h_P^*) + c\sqrt{\min\{P(\hat{h}_Q^* \neq h_P^*), \hat{P}(\hat{h}_Q^* \neq h_P^*)\}A(n_T, \delta)} + cA(n_T, \delta)$$
$$\leq c\sqrt{\varepsilon_P^{\beta_P}(\hat{h}_Q^*)A(n_T, \delta)} + cA(n_T, \delta).$$

The first inequality is due to Lemma 1 and second inequality follows from Lemma 2 and Noise assumption 1. Then we have the following result, which completes the proof:

$$\varepsilon_P(\hat{h}_Q^*) \leq cA(n_T, \delta)^{\frac{1}{2-\beta_P}}.$$

$\square$

**Lemma 2.** *If $\hat{\mathcal{R}}_P(\hat{h}_Q^*) \leq \hat{\mathcal{R}}_P(h_Q^*)$, with probability at least $1 - \delta$,*

$$\hat{\mathcal{R}}_P(\hat{h}_Q^*) - \hat{\mathcal{R}}_P(h_P^*) = \varepsilon_P(h_Q^*) + \mathcal{O}(A(n_T, \delta)).$$

*Proof.*

$$\hat{\mathcal{R}}_P(\hat{h}_Q^*) - \hat{\mathcal{R}}_P(h_P^*) \leq \hat{\mathcal{R}}_P(h_Q^*) - \hat{\mathcal{R}}_P(h_P^*)$$
$$\leq \mathcal{R}_P(h_Q^*) - \mathcal{R}_P(h_P^*) + c\sqrt{\min\{P(h_Q^* \neq h_P^*), \hat{P}(h_Q^* \neq h_P^*)\}A(n_T, \delta)} + cA(n_T, \delta)$$
$$= \varepsilon_P(h_Q^*) + \mathcal{O}(A(n_T, \delta)).$$

$\square$

**Theorem 4** (Convergence Rate for SAFARI ). *Under Assumptions 3 - 5, if $\eta_c \leq \frac{1}{4\sqrt{30}LK}$, $\eta_c = \frac{2\eta_s}{K}$,*

*and $q \leq 1/\left( \frac{4\sigma_G^2 - 4G_2(\frac{1}{2K^2} - \frac{2L\eta_s^2}{K^2})}{(1-L\eta_s)G_1} + 1 \right)$, then, the sequence $\{\mathbf{x}_r\}$ generated by SAFARI satisfies:*

$$\frac{1}{R}\sum_{r=1}^{R} \mathbb{E}\|\nabla F(\mathbf{x}_r)\|^2 \leq \frac{2(F(\mathbf{x}_0) - F(\mathbf{x}^*))}{R\eta_s} + L\eta_s(1-q)\sigma^2 + \frac{80qL^2\eta_s^2}{K}(\sigma^2 + 6K\sigma_G^2) + \frac{8Lq\eta_s}{mK}\sigma^2,$$

*where $G_1 = \min_{r\in\mathcal{T}_s} \|\nabla F(\mathbf{x}_r)\|^2$ and $G_2 = \min_{r\in\mathcal{T}_c} \left\| \frac{1}{m}\sum_{i\in[m]}\sum_{k\in[K]} \nabla F_i(\mathbf{x}_{r,k}^i) \right\|^2$.*

*Proof.* We define that there are totally $R_s = |\mathcal{T}_s| = (1-p)R$ rounds for server update, $R_c = |\mathcal{T}_c| = pR$ rounds for client update, and $R = R_s + R_c$,

When server updates:

$$\mathbb{E}_r[F(\mathbf{x}_{r+1})] \leq F(\mathbf{x}_r) + \left\langle \nabla F(\mathbf{x}_r), \mathbb{E}_r[\mathbf{x}_{r+1} - \mathbf{x}_r] \right\rangle + \frac{L}{2}\mathbb{E}_r[\|\mathbf{x}_{r+1} - \mathbf{x}_r\|^2]$$

$$= F(\mathbf{x}_r) + \left\langle \nabla F(\mathbf{x}_r), \eta_s\mathbb{E}_r[\nabla F(\mathbf{x}_r, \xi_r)] \right\rangle + \frac{L}{2}\eta_s^2\mathbb{E}_r[\|\nabla F(\mathbf{x}_r, \xi_r)\|^2]$$

$$= F(\mathbf{x}_r) - \eta_s\|\nabla F(\mathbf{x}_r)\|^2 + \frac{L\eta_s^2}{2}\|\nabla F(\mathbf{x}_r)\|^2 + \frac{L\eta_s^2}{2}\sigma^2.$$

That is,

$$\|\nabla F(\mathbf{x}_r)\|^2 \leq \frac{2}{\eta_s}(F(\mathbf{x}_r) - \mathbb{E}_r[F(\mathbf{x}_{r+1})]) + (L\eta_s - 1)\|\nabla F(\mathbf{x}_r)\|^2 + L\eta_s\sigma^2.$$

When clients update:

$$\mathbb{E}_r[F(\mathbf{x}_{r+1})] \leq F(\mathbf{x}_r) + \left\langle \nabla F(\mathbf{x}_r), \mathbb{E}_r[\mathbf{x}_{r+1} - \mathbf{x}_r] \right\rangle + \frac{L}{2}\mathbb{E}_r[\|\mathbf{x}_{r+1} - \mathbf{x}_r\|^2]$$

$$= F(\mathbf{x}_r) + \underbrace{\left\langle \nabla F(\mathbf{x}_r), -\eta_c\mathbb{E}_r[\Delta_r] \right\rangle}_{A_1} + \underbrace{\frac{L}{2}\eta_c^2\mathbb{E}_r[\|\Delta_r\|^2]}_{A_2}.$$

$$A_1 = \left\langle \nabla F(\mathbf{x}_r), -\eta_c\mathbb{E}_r[\Delta_r] \right\rangle$$

$$= \frac{1}{2K}\eta_c\left[ -K^2\|\nabla F(\mathbf{x}_r)\|^2 - \|\mathbb{E}_r[\Delta_r]\|^2 + \|K\nabla F(\mathbf{x}_r) - \mathbb{E}_r[\Delta_r]\|^2 \right]$$

$$= -\frac{K\eta_c}{2}\|\nabla F(\mathbf{x}_r)\|^2 - \frac{\eta_c}{2K}\left\| \frac{1}{m}\sum_{i\in[m]}\sum_{k\in[K]} \nabla F_i(\mathbf{x}_{r,k}^i) \right\|^2 + \frac{\eta_c}{2K}\left\| \frac{1}{m}\sum_{i\in[m]}\sum_{k\in[K]} \left[\nabla F(\mathbf{x}_r) - \nabla F_i(\mathbf{x}_{r,k}^i)\right] \right\|^2$$

$$\leq -\frac{K\eta_c}{2}\|\nabla F(\mathbf{x}_r)\|^2 - \frac{\eta_c}{2K}\left\| \frac{1}{m}\sum_{i\in[m]}\sum_{k\in[K]} \nabla F_i(\mathbf{x}_{r,k}^i) \right\|^2 + \frac{\eta_c}{2m}\sum_{i\in[m]}\sum_{k\in[K]} \underbrace{\left\|\nabla F(\mathbf{x}_r) - \nabla F_i(\mathbf{x}_{r,k}^i)\right\|^2}_{A_3}$$

$$\leq -\frac{K\eta_c}{2}\|\nabla F(\mathbf{x}_r)\|^2 - \frac{\eta_c}{2K}\left\| \frac{1}{m}\sum_{i\in[m]}\sum_{k\in[K]} \nabla F_i(\mathbf{x}_{r,k}^i) \right\|^2$$

$$+ \eta_c K\sigma_G^2 + \eta KL^2\left[5K\eta_c^2(\sigma^2 + 6K\sigma_G^2) + 30K^2\eta_c^2\|\nabla F(\mathbf{x}_r)\|^2\right]$$

$A_3$ could be bounded as follows:

$$A_3 = \left\|\nabla F(\mathbf{x}_r) - \nabla F_i(\mathbf{x}_{r,k}^i)\right\|^2$$

$$= \left\|\nabla F(\mathbf{x}_r) - \nabla F_i(\mathbf{x}_r) + \nabla F_i(\mathbf{x}_r) - \nabla F_i(\mathbf{x}_{r,k}^i)\right\|^2$$

$$\leq 2 \left\| \nabla F(\mathbf{x}_r) - \nabla F_i(\mathbf{x}_r) \right\|^2 + 2 \left\| \nabla F_i(\mathbf{x}_r) - \nabla F_i(\mathbf{x}_{r,k}^i) \right\|^2$$

$$\leq 2\sigma_G^2 + 2L^2 \left\| \mathbf{x}_r - \mathbf{x}_{r,k}^i \right\|^2$$

$$\leq 2\sigma_G^2 + 2L^2 \left[ 5K\eta_c^2(\sigma^2 + 6K\sigma_G^2) + 30K^2\eta_c^2\|\nabla F(\mathbf{x}_r)\|^2 \right],$$

where the last inequality follows from with $\eta_c \leq \frac{1}{8LK}$.

$$A_2 = \frac{L}{2}\eta_c^2 \mathbb{E}_r[\|\Delta_r\|^2]$$

$$\leq L\eta_c^2 \left\| \frac{1}{m} \sum_{i\in[m]} \sum_{k\in[K]} \nabla F_i(\mathbf{x}_{r,k}^i) \right\|^2 + L\eta_c^2 \left\| \frac{1}{m} \sum_{i\in[m]} \sum_{k\in[K]} \left[ \nabla F_i(\mathbf{x}_{r,k}^i) - \nabla F_i(\mathbf{x}_{r,k}^i, \xi_{r,k}^i) \right] \right\|^2$$

$$\leq L\eta_c^2 \left\| \frac{1}{m} \sum_{i\in[m]} \sum_{k\in[K]} \nabla F_i(\mathbf{x}_{r,k}^i) \right\|^2 + \frac{L\eta_c^2 K}{m}\sigma^2,$$

where the last inequality is due to the martingale difference sequence $\{\nabla F_i(\mathbf{x}_{r,k}^i) - \nabla F_i(\mathbf{x}_{r,k}^i, \xi_{r,k}^i)\}$ (see Lemma 4 in (Karimireddy et al., 2020)).

Putting pieces together, we have

$$K\eta_c(\frac{1}{2} - 30L^2K^2\eta_c^2)\|\nabla F(\mathbf{x}_r)\|^2 \leq F(\mathbf{x}_r) - \mathbb{E}_r[F(\mathbf{x}_{r+1}) - \frac{\eta_c}{2K} \left\| \frac{1}{m} \sum_{i\in[m]} \sum_{k\in[K]} \nabla F_i(\mathbf{x}_{r,k}^i) \right\|^2 + \eta_c K\sigma_G^2$$

$$+ \eta K L^2 \left[ 5K\eta_c^2(\sigma^2 + 6K\sigma_G^2) + 30K^2\eta_c^2\|\nabla F(\mathbf{x}_r)\|^2 \right] + L\eta_c^2 \left\| \frac{1}{m} \sum_{i\in[m]} \sum_{k\in[K]} \nabla F_i(\mathbf{x}_{r,k}^i) \right\|^2 + L\eta_c^2 K^2\sigma^2$$

If $(\frac{1}{2} - 30L^2K^2\eta_c^2) \geq \frac{1}{4}$ (i.e., $\eta_c \leq \frac{1}{4\sqrt{30}LK}$) and $\eta_c = \frac{2\eta_s}{K}$, we have

$$\|\nabla F(\mathbf{x}_r)\|^2 \leq \frac{4}{K\eta_c}(F(\mathbf{x}_r) - \mathbb{E}_r[F(\mathbf{x}_{r+1})]) + 4\sigma_G^2$$

$$+ \left(\frac{4L\eta_c}{K} - \frac{4}{2K^2}\right) \left\| \frac{1}{m} \sum_{i\in[m]} \sum_{k\in[K]} \nabla F_i(\mathbf{x}_{r,k}^i) \right\|^2 + 20KL^2\eta_c^2(\sigma^2 + 6K\sigma_G^2) + \frac{4L\eta_c}{m}\sigma^2$$

$$= \frac{2}{\eta_s}(F(\mathbf{x}_r) - \mathbb{E}_r[F(\mathbf{x}_{r+1})]) + 4\sigma_G^2$$

$$+ \left(\frac{8L\eta_s}{K^2} - \frac{4}{2K^2}\right) \left\| \frac{1}{m} \sum_{i\in[m]} \sum_{k\in[K]} \nabla F_i(\mathbf{x}_{r,k}^i) \right\|^2 + \frac{80L^2\eta_s^2}{K}(\sigma^2 + 6K\sigma_G^2) + \frac{8L\eta_s}{mK}\sigma^2$$

Note there are totally $R_s$ rounds ($T_s$ as the round indices) for server update and $R_c$ rounds ($T_c$ as the round indices) for client update. Let $R = R_s + R_c$, we have

$$\frac{1}{R}\sum_{r=1}^{R}\|\nabla F(\mathbf{x}_r)\|^2 \leq \frac{2}{\eta_s}\frac{1}{R}\sum_{r=1}^{R}(F(\mathbf{x}_r) - \mathbb{E}_r[F(\mathbf{x}_{r+1})]) + \frac{1}{R}\sum_{r\in T_s}(L\eta_s - 1)\|\nabla F(\mathbf{x}_r)\|^2 + \frac{L\eta_s R_s}{R}\sigma^2$$

$$+ \frac{4R_c}{R}\sigma_G^2 + \frac{1}{R}\sum_{r\in T_c}\left(\frac{8L\eta_s}{K^2} - \frac{4}{2K^2}\right) \left\| \frac{1}{m} \sum_{i\in[m]} \sum_{k\in[K]} \nabla F_i(\mathbf{x}_{r,k}^i) \right\|^2 + \frac{80R_cL^2\eta_s^2}{KR}(\sigma^2 + 6K\sigma_G^2) + \frac{8LR_c\eta_s}{mKR}\sigma^2$$

$$\leq \frac{2(F(\mathbf{x}_0) - F(\mathbf{x}^*))}{R\eta_s} + \frac{L\eta_s R_s}{R}\sigma^2 + \frac{80R_cL^2\eta_s^2}{KR}(\sigma^2 + 6K\sigma_G^2) + \frac{8LR_c\eta_s}{mKR}\sigma^2,$$

where the last inequality follows from

$$4R_c\sigma_G^2 \leq \sum_{r\in T_s}(1 - L\eta_s)\|\nabla F(\mathbf{x}_r)\|^2 + \sum_{r\in T_c}(\frac{4}{2K^2} - \frac{8L\eta_s}{K^2})\left\|\frac{1}{m}\sum_{i\in[m]}\sum_{k\in[K]}\nabla F_i(\mathbf{x}_{r,k}^i)\right\|^2$$

$$\leq R_s(1 - L\eta_s)G_1 + R_c(\frac{4}{2K^2} - \frac{8L\eta_s}{K^2})G_2.$$

$G_1 = \min_{r\in T_s}\|\nabla F(\mathbf{x}_r)\|^2$ and $G_2 = \min_{r\in T_c}\left\|\frac{1}{m}\sum_{i\in[m]}\sum_{k\in[K]}\nabla F_i(\mathbf{x}_{r,k}^i)\right\|^2$.

$\square$

### A.1  DISCUSSIONS

We want to cast caveats on Corollary 1. The results in Corollary 1 does not hold in *arbitrary* cases. Specifically, Corollary 1 requires both $R_s \geq \frac{4\sigma_G^2 - 4G_2(\frac{1}{2K^2} - \frac{2L\eta_s^2}{K^2})}{(1-L\eta_s)G_1}R_c$ and $R_s = \mathcal{O}(\frac{R}{mK})$. In other words, we need $\frac{4\sigma_G^2 - 4G_2(\frac{1}{2K^2} - \frac{2L\eta_s^2}{K^2})}{(1-L\eta_s)G_1} \leq \frac{R_s}{R_c} \leq \frac{c}{mK(1-\frac{c}{mK})}$, where $c$ is a constant. Approximately, $\frac{R_s}{R_c} = constant$. Due to $R_s + R_c = R$, we can see that $R_c = \Omega(R)$. So the convergence rate is $\mathcal{O}(\frac{1}{\sqrt{mKR}})$, which is the same order of $\mathcal{O}(\frac{1}{\sqrt{mKR_c}})$.

## B  EXPERIMENTS

In this section, we provide the details of the numerical experiments and some additional experimental results.

### B.1  MODELS AND DATASETS

We test the SAFARI algorithm by running two models on two different types of datasets, including 1) multinomial logistic regression (LR) on MNIST, and 2) convolutional neural network (CNN) on CIFAR-10. Both datasets are chose from a previous FL paper (McMahan et al., 2017), and they are now widely used as benchmarks for FL research (Yang et al., 2021b; Li et al., 2020b).

MNIST and CIFAR-10 have ten classes of images separately. In order to impose the heterogeneity of the data, we partition the dataset according to the number of classes ($p$) that each client contains. We distribute these data to $M = 10$ clients, and each client only has a certain number of classes. Specifically, each client randomly selects $p$ classes of images and then evenly samples training and test data-points within these $p$ classes of images without replacement. For example, if $p = 2$, each client only samples training and test data-points within two classes of images, which causes the heterogeneity among different clients. If $p = 10$, each client contains training and test samples that selects from ten classes. This situation is almost the same as i.i.d. case. Hence, the number of classes ($p$) in each client's local dataset can be used to represent the level of non-i.i.d. qualitatively. In addition, to mimic incomplete client participation, we enforce $s$ clients to be exempt from participation, where the index $s$ can be used to represent the degree of incomplete client participation. Specifically, we assume there are $M = 10$ clients in total, and $m = 5$ clients participate in each communication round. These clients are uniformly sampled from $M - s$ clients. Larger incomplete client participation index $s$ means less clients participate in the training.

For both MNIST and CIFAR-10, the learning rate is 0.1, and the local epoch is 1. For MNIST, the batch size is 64, and the total communication round is 150. For CIFAR-10, the batch size is 500, and the total communication round is 5000. To simulate the data heterogeneity, we use $p = [10, 5, 2, 1]$ as a proxy to represent the degree of non-i.i.d. on MNIST and CIFAR-10 datasets. To emulate the effect of incomplete client participation, we set $s = [0, 2, 4]$ to represent the degree of incomplete client participation for the SAFARI algorithm, the FedAvg algorithm, and the SGD algorithm. Last two algorithms are employed as the baselines to compare with our algorithm. To compare the effect of the collaboration from server, we add $[50, 100, 500, 1000]$ data to the server's side for MNIST and $[500, 5000, 10000]$ for CIFAR-10.

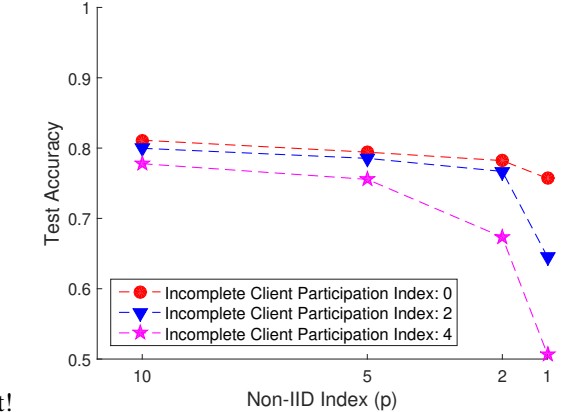

t!

Figure 4: Test Accuracy of FedAvg on CIFAR-10 with incomplete client participation. Larger incomplete client participation index means less clients participate in the training, and smaller non-i.i.d. index means the data across clients is more heterogeneous.

Table 2: Test accuracy improvement (%) for SAFARI compared with FedAvg on CIFAR-10 with incomplete client participation $s = 4$. '-' means no statistical difference within $2\%$ error bar.

| SERVER | NON-IID INDEX (P) | | | |
|---|---|---|---|---|
| DATASIZE | 10 | 5 | 2 | 1 |
| 500 | - | - | - | - |
| 5000 | - | - | 7.08 | 17.62 |
| 10000 | - | - | 9.16 | 22.65 |

## B.2 ADDITIONAL EXPERIMENTAL RESULTS

In Figure 4, we show the test accuracy of FedAvg algorithm on CIFAR-10 for different Non-IID index $p$ and incomplete client participation index $s$. In the case of $p = 10$, the test accuracy of $s = 4$ and $s = 0$ is not much different whereas the test accuracy of $s = 4$ is 25% lower than that of $s = 1$ in the case of $p = 1$. This finding on CIFAR-10 further support our first observation in Section 5. Incomplete client participation has no impact on the performance for nearly homogeneous data, but it causes catastrophical performance degradation for highly Non-IID data.

In Figure 5, we show the test accuracy of the SAFARI algorithm and the FedAvg algorithm on MNIST for incomplete client participation $s = 4$ and different Non-IID index $p$. The evidences of the observations in Section 5 are provided visually as follows:

- Compared to FedAvg in the heterogeneous case when $p = 1$ or $p = 2$ (see Figure 5(d) and 5(c)), especially when $p = 1$, with only 50 data at server's side (0.1% of the total training data), there is a non-negligible increase of test accuracy for our SAFARI algorithm. This increase increases as more data is added to the server's side.

- In nearly homogeneous case when $p = 5$ or $p = 10$ (see Figure 5(a) and 5(b)), there is actually no improvement of the test accuracy with these auxiliary data added to the server's side, comparing SAFARI with FedAvg.

In Table 2, we show the comparison between our SAFARI algorithm and FedAvg algorithm on CIFAR-10 for incomplete client participation $s = 4$. The observations in Section 5 are further illustrated: 1) There is non-negligible increase of the test accuracy for SAFARI algorithm with small amount of auxiliary data at server's side. With 10000 data at server's side, the test accuracy increases by 22.65 %. 2) There is actually no improvement with these auxiliary data for nearly homogeneous case (e.g., $p = 10$ or $p = 5$), which is denoted by '-' in the table.

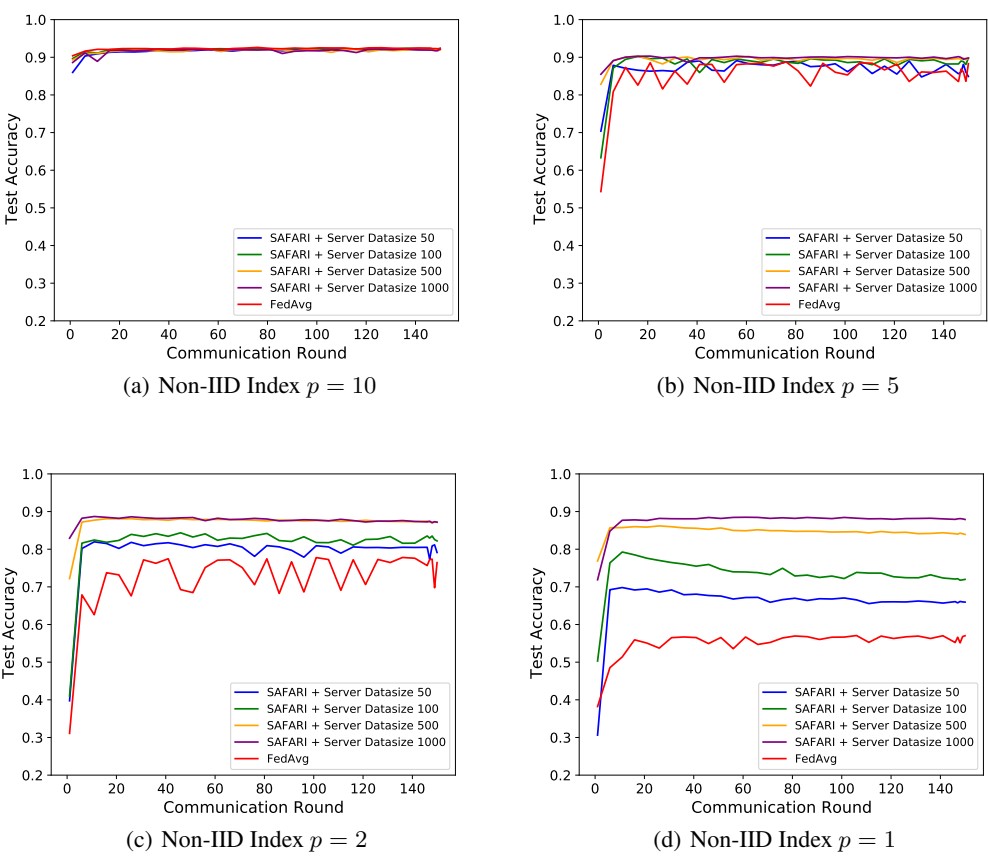

(a) Non-IID Index $p = 10$

(b) Non-IID Index $p = 5$

(c) Non-IID Index $p = 2$

(d) Non-IID Index $p = 1$

Figure 5: Test accuracy of SAFARI and FedAvg algorithm on MNIST with incomplete client participation $s = 4$ and different Non-IID index $p$. Smaller $p$ means the data across clients is more heterogeneous.