# OpenReview forum: "Understanding of  Server-Assisted Federated Learning with Incomplete Client Participation"
_ICLR.cc/2024/Conference — Submitted to ICLR 2024_

### Official Review · Reviewer_SJXE · 2023-10-29

**Soundness:** 2 fair
**Presentation:** 2 fair
**Contribution:** 2 fair
**Rating:** 5
**Confidence:** 3

**Summary:**

This work tackles data heterogeneity in federated learning, which arises from *incomplete client participation*.
The authors first analyze a general server-assisted FL framework, where the server has an auxiliary dataset to train the model.
The authors reveal that FL with incomplete client participation is not *PAC learnable*; and such an FL setting with server assistance is *PAC learnable*.
With the analysis, the authors further provide a new server-assisted FL algorithm, SAFARI, to practically address data heterogeneity.
Specifically, SAFARI has two training modes: standard FedAvg on clients and centralized learning on the server. With a small auxiliary dataset on the server, SAFARI significantly improves training performance even with some clients never participating throughout the whole training process.
The authors conduct evaluations on MNIST and CIFAR-10, which shows that SAFARI achieves much better model accuracy compared to standard FedAvg.

**Strengths:**

This paper provides a general analysis framework for server-assisted federated learning. Some contributions are highlighted below.

**1**, An analysis for FL with incomplete client participation.

I appreciate that the authors first provide a general analysis for FL with incomplete client participation. The analysis reveals why standard FL cannot address data heterogeneity with incomplete client participation. And, importantly, how server-assisted FL mitigates the problem.

**2**, A necessary clarification between SA-FL and domain adaptation.

The explanation not only discriminates SA-FL from domain adaptation methods, but also provides another perspective to understand server-assisted FL.

**3**, A new server-assisted FL algorithm with convergence guarantees.

It is good to see that the authors formally analyze the proposed algorithm with convergence guarantees.
Importantly, the analysis shows how crucial parameters such as $q$ affect convergence.

**Weaknesses:**

**1**, Regarding The Distribution of $Q$ and $P$.

Overall, I have some doubts about the distribution of the auxiliary dataset.
While there are some server-assisted FL (SA-FL) works following similar ideas, it can be too ideal to assume the server has an auxiliary dataset that has the same distribution of $P$, or even *positively-related* with P.
Usually, in standard FL, the server does not have any prior knowledge about the dataset. Such a setup comes from real concerns regarding data privacy.
However, SA-FL comes with a requirement for an auxiliary dataset that has a similar distribution as the target dataset.
It is doubtful that the server can have ways to acquire such an auxiliary dataset without breaching privacy promises.

**2**, Regarding The Analysis of FL with Incomplete Participation.

**2.1**, Definition 3 is very unclear. *Excess error* in Definition 3 is an important component in the analysis. However, it is not properly defined throughout the whole paper.

**2.2**, Theorem 1 is stated in a general way, but the proof seems a simplified case. Throughout the proof of Theorem 1, it seems a special case with only two points $x1, x2$.
As a reader, I am concerned about how it extends to general cases with an arbitrary number of points.

**2.3**, Theorem 2 is not clearly stated. First, it states Eq(1) holds with probability at least $1-\delta$. However, it is confusing how the $\delta$ is derived and how it relates to Eq(1). If $\delta$ is not in Eq(1), what is the point of having such a variable?

Another issue of Theorem 2 is that the $\alpha$ parameter in Assumption 2 is not shown up in Eq(1) either. It is strange. Imaging a very large $\alpha$ to bound the similarly of distribution $P$ and $Q$, then Theorem 2 implies that the error $\varepsilon$ is still small and is only decided by $\beta$. It does not make sense to me.

**2.4**, The assumption, $\hat{R}(\hat{h}) \leq \hat{R}(h^*)$, in Theorem 3 need further justification. As a reader, I do not see why the assumption is valid.

**3**, Regarding The Evaluation Setup.

**3.1**, For the auxiliary dataset on the server, the authors simply extract a subset of the target training dataset.
It is expected that it can give better performance compared to the training without the subset.
However, the question is, it is an unfair comparison. SAFARI allows the server to access a subset of the target training dataset, thereby breaking privacy promises in FL. It is unfair to use model accuracy as the sole criterion to compare with FedAvg.

**3.2**, The central question is again how to attain the auxiliary dataset for the server training. To have a fair comparison, I believe the authors need to find auxiliary samples that are independent of the target dataset such as other public sources or generated samples.

**Questions:**

**1**, On the top of page 6, the authors state that "Note that when $\beta \geq 1$, the first term in Eq(1) dominates". I wonder why this is the case. Do the authors assume $\frac{d_H}{n_T+n_S} \leq 1$? If so, please explain why.

---

### Official Review · Reviewer_65h6 · 2023-10-31

**Soundness:** 2 fair
**Presentation:** 3 good
**Contribution:** 2 fair
**Rating:** 3
**Confidence:** 4

**Summary:**

This paper examines the Federated Learning framework in situations where not all clients participate in the training process, potentially causing training failures. To address this issue, the Server-Assisted Federated Learning (SA-FL) approach has been introduced, which involves incorporating an additional server-side dataset. Empirical evidence has demonstrated the effectiveness of this approach. This work aims to explore the theoretical aspects of the SA-FL approach. The study reveals that the standard Federated Learning approach is not PAC-learnable in the worst-case scenario when client participation is incomplete. To address this, the authors propose the SAFARI (Server-Assisted Federated Averaging) algorithm, providing a thorough analysis of its generalization and convergence properties. The paper also includes experimental results presented in the concluding sections.

**Strengths:**

The introduction section of the paper is notably strong, as it extensively expounds upon the motivation behind the research, defines the problem under investigation, and introduces the proposed solution in a comprehensive manner.

Additionally, the paper excels in its presentation of definitions, lemmas, and theorems, ensuring that these critical components are articulated with precision and clarity, making the content accessible to a wide audience.

The connection between Server Assisted Federated Learning and Domain Adaptation is articulated with precision, contributing to a deeper understanding of the research's relevance and its place within the broader domain.

Furthermore, this paper is augmented by a series of well-executed experiments, each serving to underscore the advantages of the proposed method. These experiments not only validate the research but also offer practical insights into its real-world applicability and potential benefits.

**Weaknesses:**

**Introduction**

The introduction section lacks contemporary citations. The most recent references date back to 2021, which may be considered outdated in the dynamic field of Federated Learning. Additionally, the limited number of cited papers does not adequately reflect the current state of the field. I recommend commencing with comprehensive overview papers instead.

Wang, J., Charles, Z., Xu, Z., Joshi, G., McMahan, H. B., Al-Shedivat, M., ... & Zhu, W. (2021). A field guide to federated optimization. arXiv preprint arXiv:2107.06917.

Kairouz, P., McMahan, H. B., Avent, B., Bellet, A., Bennis, M., Bhagoji, A. N., ... & Zhao, S. (2021). Advances and open problems in federated learning. Foundations and Trends® in Machine Learning, 14(1–2), 1-210.

Then, also consider modern papers such as

Patel, K. K., Wang, L., Woodworth, B. E., Bullins, B., & Srebro, N. (2022). Towards optimal communication complexity in distributed non-convex optimization. Advances in Neural Information Processing Systems, 35, 13316-13328.

Wang, J., Lu, Y., Yuan, B., Chen, B., Liang, P., De Sa, C., ... & Zhang, C. (2023, July). CocktailSGD: fine-tuning foundation models over 500mbps networks. In International Conference on Machine Learning (pp. 36058-36076). PMLR.

Grudzień, M., Malinovsky, G., & Richtárik, P. (2023, April). Can 5th Generation Local Training Methods Support Client Sampling? Yes!. In International Conference on Artificial Intelligence and Statistics (pp. 1055-1092). PMLR.

>By carefully designing the server-client update coordination, we show that SAFARI achieves an $\mathcal{O}(1 / \sqrt{m k R})$ convergence rate to a stationary point, matching the convergence rates of state-of-the-art classic FL algorithms.

Please provide citations for this statement and mention what state-of-the-art methods and convergence rates you consider here.

**Related works**

In this section there is a lack of modern citations for Client Participation in Federated Leaning. The current overview does not cover modern methods that tackle heterogeneity:

Mishchenko, K., Malinovsky, G., Stich, S., & Richtárik, P. (2022, June). Proxskip: Yes! local gradient steps provably lead to communication acceleration! finally!. In International Conference on Machine Learning (pp. 15750-15769). PMLR.

 Mitra, A., Jaafar, R., Pappas, G. J., & Hassani, H. (2021). Linear convergence in federated learning: Tackling client heterogeneity and sparse gradients. Advances in Neural Information Processing Systems, 34, 14606-14619.

Also this section does not cover works that consider special client sampling procedures:

Malinovsky, G., Horváth, S., Burlachenko, K., & Richtárik, P. (2023). Federated learning with regularized client participation. arXiv preprint arXiv:2302.03662.

Chen, W., Horváth, S., & Richtárik, P. (2022). Optimal Client Sampling for Federated Learning. Transactions on Machine Learning Research.

Cho, Y.J., Sharma, P., Joshi, G., Xu, Z., Kale, S. &amp; Zhang, T.. (2023). On the Convergence of Federated Averaging with Cyclic Client Participation. <i>Proceedings of the 40th International Conference on Machine Learning</i>, in <i>Proceedings of Machine Learning Research</i> 202:5677-5721 Available from https://proceedings.mlr.press/v202/cho23b.html.

Moreover, related work section does not cover modern literature of asynchronous methods:

Mishchenko, K., Bach, F., Even, M., & Woodworth, B. E. (2022). Asynchronous sgd beats minibatch sgd under arbitrary delays. Advances in Neural Information Processing Systems, 35, 420-433.

Koloskova, A., Stich, S. U., & Jaggi, M. (2022). Sharper convergence guarantees for asynchronous sgd for distributed and federated learning. Advances in Neural Information Processing Systems, 35, 17202-17215.

Tyurin, A., & Richtárik, P. (2023). Optimal Time Complexities of Parallel Stochastic Optimization Methods Under a Fixed Computation Model. arXiv preprint arXiv:2305.12387.

**PAC-learnability of FL with incomplete client participation**

**Conventional Federated Leaning with Incomplete Client Participation**

This formula on page 3 might be confusing for readers:
$$\min \_{\mathbf{x} \in \mathbb{R}^d} \hat{F}(\mathbf{x})=\sum_{i \in[M]} \alpha\_i \hat{F}\_i(\mathbf{x}) \triangleq\left(1 /\left|S\_i\right|\right) \sum_{\xi \in S\_i} f\_i(\mathbf{x}, \xi).$$
It is not clear whether $  \hat{F}(\mathbf{x})=\sum_{i \in[M]} \alpha\_i \hat{F}\_i(\mathbf{x}) $ or $\hat{F}\_i(\mathbf{x}) \triangleq\left(1 /\left|S\_i\right|\right) \sum_{\xi \in S\_i} f\_i(\mathbf{x}, \xi).$ Please clarify this aspect in the text.

I carefully reviewed Theorem 1 (the Impossibility Theorem) and its corresponding proof in the appendix. The proof appears to be sound; however, I must note that I lack expertise in statistical learning theory, so I may have overlooked certain details.

It's worth noting that the provided proof heavily relies on prior work, and the outcome it yields does not appear to be particularly surprising. In essence, the core concept of this assertion revolves around the idea that in cases where a system lacks access to specific information (in this context, data from clients that are entirely uncommunicative), it is unfeasible to construct a model with an error rate tending towards zero. This seems to be a rather self-evident conclusion, especially when considering scenarios where such non-communicative clients possess unique and isolated data that does not intersect with the data available from other clients. In such cases, it becomes impractical to derive a model suitable for the broader data distribution.

With all due respect, the term "fundamental" ascribed to this result, as well as the subsequent assertion that "This result sheds light on system and algorithm design for FL," might be perceived as somewhat exaggerated and overstated.

> In addition to system heterogeneity, other factors such as Byzantine attackers could also render incomplete client participation. For example, even for full client participation in FL, if part of the clients are Byzantine attackers, the impossibility theorem also applies.

I respectfully disagree with this statement. In the context of Byzantine robustness, we follow a standard formulation prevalent in the literature on Byzantine robustness, as evidenced by the following references:

Karimireddy, S. P., He, L., & Jaggi, M. (2021, October). Byzantine-Robust Learning on Heterogeneous Datasets via Bucketing. In International Conference on Learning Representations.

Gorbunov, E., Horváth, S., Richtárik, P., & Gidel, G. (2022). Variance reduction is an antidote to byzantines: Better rates, weaker assumptions and communication compression as a cherry on the top. arXiv preprint arXiv:2206.00529.

We assume that there are $n$ clients consisting of the two groups: $[n]=\mathcal{G} \sqcup \mathcal{B}$, where $\mathcal{G}$ denotes the set of good clients and $\mathcal{B}$ is the set of bad/malicious/Byzantine workers. The goal is to solve the following optimization problem
$$
\min \_{x \in \mathbb{R}^d} f(x)=\frac{1}{G}  \sum\_{i \in \mathcal{G}} f\_i(x),$$
where $G=|\mathcal{G}|$. In essence, we seek to build a model solely based on data from the good clients. Therefore, we do not factor in data from Byzantine workers in our problem. As a result, the concept of incomplete client participation is not applicable to this particular scenario. If there are any aspects I may have missed, please clarify.

** The PAC Learnability of Server-Assisted Federated Learning**

>The intuition of SA-FL is to utilize a dataset $T$ i.i.d. sampled from distribution $P$ with cardinality $|T|=n_T$ as a vehicle to correct potential distribution deviations due to incomplete client participation. By doing so, the server steers the learning by a small number of representative data, while the clients assist the learning by federation to leverage the huge amount of privately decentralized data $\left(n_S \gg n_T\right)$. Note that the assumption of having a server-side dataset is not restrictive since such datasets are already available in many FL systems: although not always necessary for training, an auxiliary dataset is often needed for defining FL tasks (e.g., simulation prototyping) before training and model checking after training (e.g., quality evaluation and sanity checking) (McMahan et al., 2021; Wang et al., 2021a).

I concur that in practical applications, the existence of server-side datasets is plausible. Nevertheless, I find it challenging to accept the assumption that the server-side dataset T is independently and identically distributed (i.i.d) from the distribution P without limitations. I firmly believe that if the server-side dataset is derived from distribution P in any manner, it implies that the server must possess some level of access to data from all clients to obtain it. Gaining access to clients' raw data stands in direct opposition to the fundamental principles of Federated Learning and constitutes a severe breach of privacy. Otherwise, inherent biases may arise in the dataset. In practice, achieving a server-side dataset that is i.i.d. sampled from distribution P is often unattainable, and it may be more reasonable to consider alternative distributions or methods for constructing server-side data.

Could you kindly provide more detailed information regarding this matter?

>Assumption 2 specifies a stronger constraint between distributions $P$ and $Q$. It implies that the difference of excess error for one hypothesis $h \in \mathcal{H}$ between $P$ and $Q$ is bounded by the excess error of $Q$ in some exponential form. Assumption 2 is one of the major novelty in our paper and unseen in the literature. We note that this $(\alpha, \beta)$-positively-related condition is a mild condition.

Could you please provide a more comprehensive explanation as to why this assumption is necessary in the analysis and elucidate its role?

Upon reviewing the proof for Theorem 3, it becomes apparent that it heavily relies on the assumption that the dataset T is independently and identically distributed (i.i.d) from the distribution P. I would appreciate it if you could clarify the outcomes and implications in situations where the server-side dataset T is sourced from a non-i.i.d or biased distribution.

>Last but not least, it is worth pointing out that, for ease of illustration, Theorem $2-3$ are based on the assumption that the auxiliary dataset $T \stackrel{i . i . d .}{\sim} P$. Nonetheless, it is of practical importance to consider the scenario where $T$ is sampled from a related but slightly different distribution $P^{\prime}$ rather than the target distribution $P$ itself. In fact, the above assumption could be relaxed to $T \stackrel{i . i . d .}{\sim} P^{\prime}$ for any $P^{\prime}$ as long as the mixture distribution $Q=\lambda_1 D+\lambda_2 P^{\prime}$ is $(\alpha, \beta)$-positively-related with $P$. Under such condition, we can show that the main results in Theorem 2-3 continue to hold.

Can you please characterize the difference between distribution $P$ and $P^{\prime}$ such that mixture distribution $Q=\lambda_1 D+\lambda_2 P^{\prime}$ is $(\alpha, \beta)$-positively-related with $P$?

**THE SAFARI ALGORITHM FOR TRAINING UNDER SA-FL**

Assumption 5 appears to be rather restrictive, as it necessitates the bound to hold for all $i$ in the range $[M]$, and the overall expression is constrained by a constant. I would like to propose a more general assumption:

$$\frac{1}{G} \sum\_{i \in \mathcal{G}}\left\Vert\nabla f\_i(x)-\nabla f(x)\right\Vert^2 \leq B\Vert\nabla f(x)\Vert^2+\zeta^2 \quad \forall x \in \mathbb{R}^d.$$

It's worth noting that this assumption is a standard concept well-documented in the existing literature.

Karimireddy, S. P., Kale, S., Mohri, M., Reddi, S., Stich, S., & Suresh, A. T. (2020, November). Scaffold: Stochastic controlled averaging for federated learning. In International conference on machine learning (pp. 5132-5143). PMLR.

Gorbunov, E., Horváth, S., Richtárik, P., & Gidel, G. (2022). Variance reduction is an antidote to byzantines: Better rates, weaker assumptions and communication compression as a cherry on the top. arXiv preprint arXiv:2206.00529.


I have significant concerns regarding Theorem 4. Firstly, the constants, where $G_1=\min_{r\in\mathcal{T}s}\left\Vert\nabla F(\mathbf{x}r)\right\Vert^2$ and $G_2=\min{r\in\mathcal{T}c}\left\Vert\frac{1}{m}\sum{i\in[m]}\sum{k\in[K]}\nabla F_i(\mathbf{x}_{r,k}^i)\right\Vert^2$, do not provide any information regarding their potential magnitudes. These constants necessitate the calculation of the minimum among server iterates for $G_1$ and the minimum among client iterates for $G_2$. Including such terms in the convergence bound is impractical, both in theory and practice, as estimating these constants is infeasible.

Upon reviewing the proof of Theorem 4, I observed that the analysis was conducted in a less rigorous manner. It involved bounding sums of client gradients by the minimum of such sums. Similar analytical approaches were utilized in subsequent works. In this works such sum as $\left\Vert\frac{1}{m} \sum_{i \in[m]} \sum_{k \in[K]} \nabla F_i\left(\mathbf{x}_{r, k}^i\right)\right\Vert^2 $ are bounded in more rigorous way. Please clarify this aspect of the proof.

Moreover, the end of proof is not correct:
$$
4 R\_c \sigma\_G^2  \leq \sum\_{r \in T\_s}\left(1-L \eta\_s\right)\left\Vert\nabla F\left(\mathbf{x}\_r\right)\right\Vert^2+\sum_{r \in T\_c}\left(\frac{4}{2 K^2}-\frac{8 L \eta_s}{K^2}\right)\left\Vert\frac{1}{m} \sum_{i \in[m]} \sum\_{k \in[K]} \nabla F_i\left(\mathbf{x}\_{r, k}^i\right)\right\Vert^2  \leq R\_s\left(1-L \eta_s\right) G\_1+R_c\left(\frac{4}{2 K^2}-\frac{8 L \eta\_s}{K^2}\right) G\_2,
$$
where $G_1=\min_{r\in\mathcal{T}s}\left\Vert\nabla F(\mathbf{x}r)\right\Vert^2$ and $G_2=\min{r\in\mathcal{T}c}\left\Vert\frac{1}{m}\sum{i\in[m]}\sum{k\in[K]}\nabla F_i(\mathbf{x}_{r,k}^i)\right\Vert^2$

The term "minimum" should be replaced with "maximum" since the formula requires an upper bound. I acknowledge that this is likely a typographical error, but the criticism mentioned earlier still holds true.

It's also unclear why bounding the term $\frac{1}{R} \sum_{r=1}^R \mathbb{E}\left\Vert\nabla F\left(\mathbf{x}_r\right)\right\Vert^2$ is meaningful and what it signifies. Typically, in the non-convex literature, the term $\min _{0 \leq k \leq K-1} \mathbb{E}\left[\left\Vert\nabla f\left(x_k\right)\right\Vert^2\right]$ is utilized in the bound.

Karimireddy, S. P., Kale, S., Mohri, M., Reddi, S., Stich, S., & Suresh, A. T. (2020, November). Scaffold: Stochastic controlled averaging for federated learning. In International conference on machine learning (pp. 5132-5143). PMLR.

Khaled, A., Mishchenko, K., & Richtárik, P. (2020, June). Tighter theory for local SGD on identical and heterogeneous data. In International Conference on Artificial Intelligence and Statistics (pp. 4519-4529). PMLR.

Malinovsky, G., Mishchenko, K., & Richtárik, P. (2022). Server-side stepsizes and sampling without replacement provably help in federated optimization. arXiv preprint arXiv:2201.11066.

The most recent paper, titled "Server-side stepsizes and sampling without replacement provably help in federated optimization," delves into an algorithm closely related to the one discussed in this study, and their analytical approach bears similarities. However, it notably avoids the pitfalls associated with bounding sums by the minimum among server and client iterates.

I would suggest conducting a comparative analysis of the results obtained in this study with those derived from the aforementioned relevant papers to gain a deeper understanding of the strengths and limitations of each approach.

>Further, Theorem 4 immediately implies that, by choosing parameters $q$ and the learning rate $\eta$ appropriately, we achieve linear convergence speedup to a stationary point:

What is $\eta$ in this sentence? Is this server or client stepsize?

**Numerical experiments**

In this section, there is a notable absence of a comparative analysis with respect to baselines other than FedAvg. To offer a more comprehensive evaluation, it would be advantageous to include a comparison with additional baseline algorithms commonly employed in similar studies. This would provide a more holistic perspective on the performance of the proposed approach.

Furthermore, to enhance the clarity and effectiveness of the analysis, it is advisable to incorporate graphical representations such as plots depicting the convergence of the algorithm in terms of the loss function. Visualizing the convergence process can aid in conveying a more intuitive understanding of the algorithm's performance and highlight any distinct advantages or shortcomings it may exhibit in relation to other baselines.

**Questions:**

Please answer the questions I mentioned in the section "Weaknesses".

Also I would like to ask additional questions:

How does the convergence rate depend on size of auxiliary server-side dataset?

How does the auxiliary server-side dataset change the ERM problem?

---

### Official Review · Reviewer_JHPN · 2023-11-10

**Soundness:** 3 good
**Presentation:** 2 fair
**Contribution:** 2 fair
**Rating:** 3
**Confidence:** 4

**Summary:**

This paper investigates the generalization error in conventional federated learning (FL) scenarios where client participation is incomplete, utilizing the framework of Probabilistically Approximately Correct (PAC) learnability. To effectively tackle the challenges posed by incomplete client participation, this paper proposes SAFARI—a novel framework designed to ensure both convergence and communication efficiency.

**Strengths:**

1. This paper focuses on an interesting problem. In FL, there are often constraints like bandwidth limitations or delays, preventing clients from fully participating in training. This can lead to convergence problems or introduce additional generalization errors.
2. The writing is relatively clear and understandable.
3. There is a certain theoretical guarantee.

**Weaknesses:**

1. The proposed framework relies on strong assumptions, such as the need for the server to hold certain representative samples. However, this may contradict the privacy assumptions of federated learning.
2. The experiments are not very comprehensive, especially in terms of the baselines compared.
3. There are some grammatical errors.

**Questions:**

1. This paper claims that no algorithm can outperform random guessing. I am somewhat puzzled by this. Theorem 1 indicates that for any learning algorithm, there exists a distribution $P$ such that the probability of learning failure is significant ($p>0.05$). However, I wonder if this necessarily implies that no algorithm can learn a better predictor than random guessing.
2. The framework here requires the server to collect samples that accurately reflect the underlying distribution of clients, which is usually considered impractical in federated settings, whether for privacy or communication reasons.
3. Overall, this paper aims to address the challenge of generalization in FL . In the related work section, the authors review numerous relevant studies, including various domain adaptation techniques in FL. However, these methods or baselines seem to be mysteriously missing in the experiments.
4. In the experiment, there are a total of $M = 10$ clients, with $m = 5$ participating in each communication round. This means that $50%$ of the clients are selected in each round, which seems inconsistent with the background introduced by this paper.
5. How is the value of $q$ determined in the experiment? Is there any theoretical basis for this choice?
6. Some grammatical errors：

In abstract, ‘Existing works in federated learning (FL) often assumes’ —> ‘Existing works… **assume**’;

‘To mitigate impacts of incomplete client participation, a popular … ‘ —> ‘A popular approach to mitigate **the impacts** of incomplete client participation is the server-assisted federated learning (SA-FL) framework….’;

‘there remains a lack of theoretical understanding for SA-FL’ —> ‘there remains a lack of theoretical understanding **of** SA-FL’;

‘Meanwhile, the ramifications of incomplete client participation in conventional FL is also poorly understood.’ —> ‘the ramifications… **are** ’;

‘These theoretical gaps motivate us to rigorously investigate SA-FL.’ —> ‘These theoretical gaps motivate us to investigate SA-FL **rigorously**’;

‘Extensive experiments on different datasets show SAFARI significantly improve the performance under incomplete client participation.’ —> ‘SAFARI significantly **improves**’.

---

### Official Review · Reviewer_Yuno · 2023-11-15

**Soundness:** 3 good
**Presentation:** 4 excellent
**Contribution:** 3 good
**Rating:** 8
**Confidence:** 3

**Summary:**

The paper is about dealing with partial participation in federated learning. Indeed, in the heterogeneous setting, there is a drift in the obtained solution if not all clients participate. the authors study this discrepancy and propose new algorithms to mitigate it. They consider the idea of sever-aided federated learning (SA-FL), which is to equip the server with a small auxiliary dataset that approximately mimics the population distribution. They provide new results on SA-FL and propose a new algorithm, called SAFARI, which handles partial participation and is communication-efficient.

I reviewed a previous version of this paper with title "ON THE EFFICACY OF SERVER-AIDED FEDERATED LEARNING AGAINST PARTIAL CLIENT PARTICIPATION" last year, which has been rejected. I again recommend acceptance, since the results are interesting, timely, and, as far as I can tell, new.

I did not check the correctness of the theorems, as several notions from statistics are too far away from my competence in optimization.

**Strengths:**

The paper provides interesting theoretical insights about heterogeneity and the important challenge of partial participation in FL.

**Weaknesses:**

Assumption 5 on Bounded Gradient Dissimilarity is restrictive. In particular, it does not apply to strongly convex functions, unless they differ only by a linear term.


When mentioning works where partial participation is modeled by uniform random sampling, I recommend to add the 2 following recent papers:
* M Grudzień, G Malinovsky, P Richtárik, "Can 5th Generation Local Training Methods Support Client Sampling? Yes!",
International Conference on Artificial Intelligence and Statistics,  pp. 1055-1092, 2022
* L. Condat, I. Agarský, G. Malinovsky, and P. Richtárik, "TAMUNA: Doubly Accelerated Federated Learning with Local Training, Compression, and Partial Participation," preprint arXiv:2302.09832, 2023.

Typos:
* infinite many samples -> infinitely many samples
* Assumption 3: there is "partial" at the end

**Questions:**

Can linear convergence be proved for strongly convex functions? This would be a useful result.

---

### Meta-Review · Area_Chair_4ApY · 2023-12-23

**Metareview:**

For parts (a) and (b), I take the liberty to use a (slightly modified) summary of one of the reviewers, as it is quite extensive and appropriate:

(a)

This paper examines the Federated Learning framework in situations where not all clients participate in the training process, potentially causing training failures. To address this issue, the Server-Assisted Federated Learning (SA-FL) approach has been introduced, which involves incorporating an additional server-side dataset. Empirical evidence has demonstrated the effectiveness of this approach. This work aims to explore the theoretical aspects of the SA-FL approach. The study reveals that the standard Federated Learning approach is not PAC-learnable in the worst-case scenario when client participation is incomplete. To address this, the authors propose the SAFARI (Server-Assisted Federated Averaging) algorithm, providing a thorough analysis of its generalization and convergence properties. The paper also includes experimental results presented in the concluding sections.

(b)

The introduction section of the paper extensively covers motivation, defines the problem, and introduces the proposed solution. The paper excels in its presentation of definitions, lemmas, and theorems, ensuring that these critical components are articulated with precision and clarity, making the content accessible to a wide audience. The connection between Server Assisted Federated Learning and Domain Adaptation is articulated with precision, contributing to a deeper understanding of the research's relevance and its place within the broader domain. Furthermore, this paper is augmented by a series of well-executed experiments, each serving to underscore the advantages of the proposed method. These experiments not only validate the research but also offer practical insights into its real-world applicability and potential benefits.

(c) There are many weaknesses mentioned by all reviewers, including:

Assumption 5 on Bounded Gradient Dissimilarity is restrictive. In particular, it does not apply to strongly convex functions, unless they differ only by a linear term.

- When mentioning works where partial participation is modeled by uniform random sampling, it is recommended to add the 2 following recent papers: M Grudzień, G Malinovsky, P Richtárik, "Can 5th Generation Local Training Methods Support Client Sampling? Yes!", International Conference on Artificial Intelligence and Statistics, pp. 1055-1092, 2022;  L. Condat, I. Agarský, G. Malinovsky, and P. Richtárik, "TAMUNA: Doubly Accelerated Federated Learning with Local Training, Compression, and Partial Participation," preprint arXiv:2302.09832, 2023.
- The proposed framework relies on strong assumptions, such as the need for the server to hold certain representative samples. However, this may contradict the privacy assumptions of federated learning.
- The experiments are not very comprehensive, especially in terms of the baselines compared.
- Reviewer 65h6 mentioned a large number of weaknesses in detail; including issues with some of the mathematics
- Reviewer SJXE also listed a large number of weaknesses


---

I am very sorry to see that no rebuttal was posted in this case.

**Justification For Why Not Higher Score:**

The authors did not post any rebuttal; and hence all issues listed remained unaddressed.

**Justification For Why Not Lower Score:**

N/A

---

### Decision · Program_Chairs · 2024-01-16

Reject